# Translation elicits a growth rate-dependent, genome-wide, differential protein production in *Bacillus subtilis*

Olivier Borkowski[1,2], Anne Goelzer[2], Marc Schaffer[3], Magali Calabre[1], Ulrike Mäder[3], Stéphane Aymerich[1], Matthieu Jules[1,*] & Vincent Fromion[2,**]

## Abstract

Complex regulatory programs control cell adaptation to environmental changes by setting condition-specific proteomes. In balanced growth, bacterial protein abundances depend on the dilution rate, transcript abundances and transcript-specific translation efficiencies. We revisited the current theory claiming the invariance of bacterial translation efficiency. By integrating genome-wide transcriptome datasets and datasets from a library of synthetic *gfp*-reporter fusions, we demonstrated that translation efficiencies in *Bacillus subtilis* decreased up to fourfold from slow to fast growth. The translation initiation regions elicited a growth rate-dependent, differential production of proteins without regulators, hence revealing a unique, hard-coded, growth rate-dependent mode of regulation. We combined model-based data analyses of transcript and protein abundances genome-wide and revealed that this global regulation is extensively used in *B. subtilis*. We eventually developed a knowledge-based, three-step translation initiation model, experimentally challenged the model predictions and proposed that a growth rate-dependent drop in *free* ribosome abundance accounted for the differential protein production.

**Keywords** *Bacillus subtilis*; global regulation; growth rate; protein production; translation efficiency

**Subject Categories** Genome-Scale & Integrative Biology; Quantitative Biology & Dynamical Systems; Protein Biosynthesis & Quality Control

**Mol Syst Biol. (2016) 12: 870**

## Introduction

The physiological state of the cell results from a complex interplay between environmental stimuli and the molecular mechanisms generating the major cellular functions (Crick, 1970; Bollenbach *et al*, 2009; Scott *et al*, 2010). As a result, each environmental condition coincides with a specific growth rate and a growth rate-dependent macromolecular composition (Schaechter *et al*, 1958; Bremer & Dennis, 2008). The abundance of the molecular machines (DNA and RNA polymerases, ribosome) strongly increases with increasing growth rate (Bremer & Dennis, 2008; Klumpp & Hwa, 2008; Klumpp *et al*, 2009). The pool of *free* polymerase (*i.e.* RNAP$\sigma^{70}$ available to initiate transcription) as well as the overall transcription efficiency increases, which in turn leads to a significant increase in total RNA and total rRNA abundances (Schaechter *et al*, 1958; Maaløe & Kjeldgaard, 1966; Marr, 1991; Bremer & Dennis, 2008; Klumpp & Hwa, 2008). In addition, the global growth rate-dependent variation in the transcription machinery abundance strongly and specifically influences the expression of each gene on the basis of their promoter sequence (Klumpp & Hwa, 2008; Gerosa *et al*, 2013). This effect is referred to as "global regulation", and the promoter activity can be described by a Michaelis–Menten-type rate law as a function of the growth rate (Gerosa *et al*, 2013).

Global regulation operates at the level of translation by identically altering the production of each protein due the growth-related dilution (Liang *et al*, 2000; Bremer & Dennis, 2008; Klumpp *et al*, 2009). In contrast to transcription, the global growth rate-dependent variation in the translation machinery does not seem to trigger an additional global regulation specific to the genetic sequence of the translation initiation region (TIR). Indeed, the transcript-specific translation efficiency, defined as the number of proteins produced per mRNA per hour, was estimated to be invariant (Bremer & Dennis, 2008). As a consequence, the translation efficiency ($\lambda_i$) of each transcript ($m_i$) can be described by a constant (*i.e.* leading to the protein abundance: $P_i = \frac{m_i}{\mu} \lambda_i$ with $\mu$ being the rate of growth in $h^{-1}$ (Klumpp *et al*, 2009)).

The advent of high-resolution technologies and the consecutive generation of quantitative, genome-wide transcriptomic and proteomic datasets (Nicolas *et al*, 2012; Muntel *et al*, 2014; Goelzer *et al*, 2015) enable the estimation of the transcript-specific translation efficiencies genome-wide. The comparison of these proteome

1 Micalis Institute, INRA, AgroParisTech, Université Paris-Saclay, Jouy-en-Josas, F78350, France
2 MaIAGE, INRA, Université Paris-Saclay, Jouy-en-Josas, F78350, France
3 Interfaculty Institute for Genetics and Functional Genomics, University Medicine Greifswald, Greifswald, Germany
  *Corresponding author. Tel: +33 134652956; E-mail: Matthieu.Jules@grignon.inra.fr
  **Corresponding author. Tel: +33 134652881; E-mail: vincent.fromion@jouy.inra.fr

and transcriptome datasets immediately suggests that the current model of the invariant transcript-specific translation efficiency is not satisfactory at the genome-scale level. The aim of this work was therefore to investigate the transcript-specific translation efficiency across growth conditions and to revisit the global regulation operating at the translation level. To this purpose, we combined genome-wide arrays, proteomics, qPCR and fluorescent reporter fusions to quantify transcript and protein abundances and deduce the transcript-specific translation efficiencies across growth conditions. We first demonstrated that translation efficiency does not remain constant but drops when growth rate increases. We furthermore revealed that the gene-specific translation initiation region (TIR) can drive a differential production of single proteins in the absence of any dedicated, specific regulators. We showed that the transcript-specific translation efficiency can be described by a Michaelis–Menten-type rate law as a function of the *free* ribosome abundance. We proposed that the drop in translation efficiency can result from a drop in abundance of the *free* ribosomes with increasing growth rates and estimated the drop in *free* ribosome abundance as well as the parameters of the model for over a thousand of transcripts. We further explored the sensitivity of the translational global regulation with respect to the addition of translation inhibitors. To precisely investigate the growth-rate dependency of the translation efficiency, we eventually developed a knowledge-based mathematical model of protein production and explored the possible interdependence of *free* ribosome, total ribosome and mRNA abundances.

# Results

### Growth-rate dependence of bacterial transcript abundances

We determined the intracellular abundance (per mass) of total RNA species in *B. subtilis* with increasing growth rate (μ). We extracted total RNA from cells grown at various growth rates (from μ = 0.25 to 1.70 h$^{-1}$), measured the amount of total RNA and the total amount of ribosomal RNA species and inferred the corresponding abundances (see Appendix section 1.3). Total RNA abundance increased as fast as the growth rate increased (Figs 1A and EV1A). The proportion of rRNA in total RNA remained constant at ~85% (Fig EV1B–D). We next developed a dedicated experimental approach to quantify the proportion of total mRNA in total RNA for each sample using genome-wide expression microarrays and a set of control "spike-in" transcripts (Materials and Methods, Fig 1B, Appendix section 1.4). A scaling factor was inferred from the intensity values of the retrotranscribed, *in vitro*-synthesized transcripts for each microarray (Fig 1C). The result was that the proportion of total mRNA in total RNA remained constant across samples, which directly implies that total mRNA abundance in *B. subtilis* does increase proportionally to the growth rate (Fig 1D).

By comparing the transcriptomes acquired in slow vs. fast growth conditions (Fig EV1E and F), we sorted the transcripts into two groups (Fig 1E): transcripts whose abundances increased faster than the total mRNA abundance (enriched species) and the remaining ones (depleted species). The first group represented ~39% of the total mRNA abundance at slow growth and reached up to 81% at fast growth. In particular, transcripts coding for ribosomal proteins

that belong to the first group increased 2.5-fold from slow to fast growth (ribosomal mRNAs). Several unregulated (constitutive) genes fell into either of the two groups (Fig EV1G). Taken together, these results reveal a global, growth rate-dependent reorganization of the transcriptome (Fig 1E) and are consistent with the global regulation of the transcription machinery (Bremer & Dennis, 2008; Klumpp & Hwa, 2008; Gerosa *et al*, 2013).

### Growth-rate dependence of the bacterial translation efficiency

In this work, we aimed at experimentally determining the transcript-specific translation efficiency across growth conditions. The transcript-specific translation efficiency ($\lambda_i$ in h$^{-1}$) is defined as the number of proteins produced per mRNA per hour, that is $\lambda_i = \frac{\mu P_i}{m_i}$ (Bremer & Dennis, 2008). In order to accurately estimate the translation efficiency, we determined the abundance of a the stable variant of the green fluorescent protein (GFPmut3 (Botella *et al*, 2010)) at various growth rates in *B. subtilis* strains carrying *gfpmut3* under the control of the promoter and translation initiation region (TIR) of the constitutive *fbaA* gene at the *fbaA* genomic locus ($^{fbaA}$TIR$_{fbaA}$ *gfp*, Table 1). GFP abundance from the $^{fbaA}$TIR$_{fba}$ *gfp*-reporter strain exhibited a fivefold decline from slow to fast growth (Fig 2A). In order to systematically correct for possible differential stability between the $^{fbaA}$TIR$_{fbaA}$ *gfp* mRNA and the *fbaA* mRNA (Fig 2B) across growth conditions, we assessed the growth rate-dependent variation in $^{fbaA}$TIR$_{fbaA}$ *gfp* mRNA abundance by quantitative PCR (qPCR, Appendix section 1.5). The $^{fbaA}$TIR$_{fbaA}$ *gfp* transcript was slightly less stable than the *fbaA* transcript, in a growth rate-dependent manner (Fig 2C). Combining total mRNA quantification and qPCR-corrected transcriptome data, we then deduced that the translation efficiency of the $^{fbaA}$TIR$_{fbaA}$ *gfp* construct decreased fourfold from slow to fast growth (Fig 2D). We concluded that the resulting protein abundance decreased with increasing growth rate (from 0.4 to 1.7 h$^{-1}$), due equally to growth-related dilution (~fourfold) and to a decrease in the growth rate-dependent translation efficiency (~fourfold).

### Translation initiation regions trigger different translation efficiency's growth-rate dependencies, and consequently differential growth rate-dependent protein production

Since the translation efficiency of a given transcript also depends on the genetic sequence of its translation initiation region (TIR (Vellanoweth & Rabinowitz, 1992)), we wondered whether the same constitutive protein expressed under the control of different TIRs may exhibit a variety of growth-rate dependencies. We therefore constructed a series of *gfp*-reporter strains combining one of the two promoters, P$_{fbaA}$ and P$_{hs}$ (a synthetic, isopropyl β-D-1-thio-galactopyranoside (IPTG)-inducible promoter) with one of eight additional synthetic translation initiation regions (TIRs, Table 1). The synthetic TIRs were derived from the natural TIR of *fbaA* (TIR$_{fbaA}$) by introducing point mutations into the RBS, the accommodation region of the ribosome, and/or changing both the sequence and the size of the accommodation region and modifying the start codon (Table 1). The synthetic TIRs were then inserted downstream of either the promoter and 5′UTR of *fbaA* ($^{fbaA}$TIR) or the promoter and 5′UTR of the artificial P$_{hs}$ ($^{hs}$TIR). For each synthetic strain, mRNAs were extracted from cells grown at

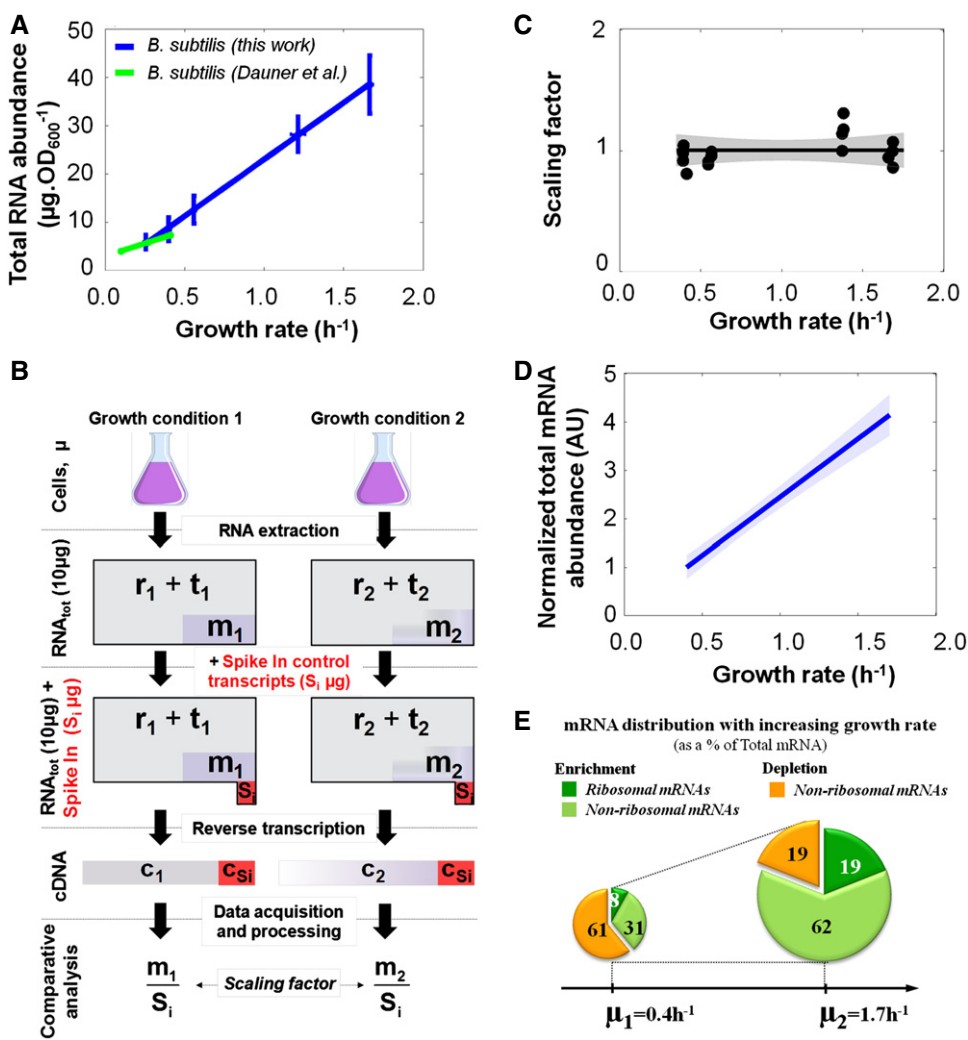

**Figure 1. Total RNA and mRNA abundances increase proportionally to the growth rate.**

A   Total RNA abundance in *B. subtilis* (blue dots) at different growth rates (0.25 $h^{-1}$, 0.40 $h^{-1}$, 0.60 $h^{-1}$, 1.2 $h^{-1}$ and 1.7 $h^{-1}$; this work). The 95% confidence intervals are shown with horizontal and vertical bars. Previously quantified total RNA abundance in *B. subtilis* (green dots) from Dauner *et al* (2001). Data are fit with a first-order polynomial.

B   Scheme of the experimental design developed to quantify the relative abundance of total mRNA and of each transcript across growth conditions. $r_1$ and $r_2$ stand for total ribosomal RNA amounts, $t_1$ and $t_2$ stand for total transfer RNA amounts, and $m_1$ and $m_2$ stand for total messenger RNA amounts in growth conditions 1 and 2, respectively. $S_i$ corresponds to the amount of the ten different *in vitro*-synthesized (spike-in) transcripts. $c_1$, $c_2$ and $c_{si}$ correspond to the amounts of cDNA reverse transcribed from $m_1$, $m_2$ and $S_i$. Note that the amount of $S_i$ added to the total RNA extract of either growth condition is identical, which straightforwardly allows calculating the scaling factor between growth conditions (see Appendix section 1.4).

C   Mean-normalized scaling factor.

D   First-order polynomial fit of normalized total mRNA abundance (as inferred from the scaling factor and total RNA abundance). The total mRNA abundance measured at slow growth ($\mu = 0.4$ $h^{-1}$) was then assigned a value of 1.0 and total mRNA abundances across the other conditions were reported normalized to this value (in arbitrary units, AU; note that AU is a quantity.$OD_{600}^{-1}$). The 95% confidence interval is shown with shaded areas.

E   The population of transcripts was sorted into two groups using a slow ($\mu = 0.4$ $h^{-1}$) and a fast growth condition ($\mu = 1.7$ $h^{-1}$). The first group contained transcripts for which the abundance increased faster than the growth rate (green), and the second group included all other transcripts (orange). The enriched sub-group of transcripts at fast growth contained the ribosomal mRNAs (dark green) and several non-ribosomal mRNAs (light green). The corresponding datasets can be found in Dataset EV1.

various growth rates and qPCRs were performed to correct for the stability differences between the various synthetic transcripts (Appendix section 1.5; Fig EV2A–D). We next quantified in media supporting growth rates ranging from 0.25 to 1.70 $h^{-1}$ the GFP abundances of 6 strains, which exhibited significantly dissimilar gene expression profiles with respect to the growth rate

($^{fbaA}$TIR$_{fbaA}$ *gfp*, $^{fbaA}$TIR$_{short}$ *gfp*, $^{fbaA}$TIR$_0$ *gfp*, $^{fbaA}$TIR$_{modif1}$ *gfp*, $^{fbaA}$TIR$_{modif2}$ *gfp* and $^{hs}$TIR$_0$ *gfp*; Figs 3A and EV2E and F, Appendix Table S1, Dataset EV2) and computed the ratios of the TIR-specific translation efficiencies. Hence, we performed minimization of mean-square-error-based estimation of the ratios of the translation efficiencies for every pair of strains (12 technical replicates *2

**Table 1. Strains and translation initiation regions (TIRs) used in this study.**

| Strains* | | Genetic sequence upstream of the reporter gene (if relevant) | | | | | |
| Name¶ | Explicit naming $^{Promoter}$TIR$_{5'UTR}$ *reporter* | Promoter | Upstream 5′ UTR† | Pre-sequence | RBS‡ | AR§ | IC‖ |
|---|---|---|---|---|---|---|---|
| OB01 | – | – | – | – | – | – | – |
| OB02 | $^{fbaA}$TIR$_0$ *gfp* | *fbaA* | GGA…88bp…ACA | G GTG GGA | AGG AGG | TGA TCC A | ATG |
| OB11 | $^{fbaA}$TIR$_{short}$ *gfp* | *fbaA* | GGA…88bp…ACA | G GTG GGA | AGG AGG | AAC TAC T | ATG |
| OB04 | $^{fbaA}$TIR$_{fbaA}$ *gfp* | *fbaA* | GGA…88bp…ACA | G GTG GGA | AGG AGG | ACA TTC GAC | ATG |
| OB06 | $^{fbaA}$TIR$_4$ *gfp* | *fbaA* | GGA…88bp…ACA | G GTG GGA | AGG AGG | ACA TTC GAC | GTG |
| OB10 | $^{fbaA}$TIR$_8$ *gfp* | *fbaA* | GGA…88bp…ACA | G GTG GGA | AGG AGG | GGG TTC GAC | ATG |
| OB03 | $^{fbaA}$TIR$_{modif1}$ *gfp* | *fbaA* | GGA…88bp…ACA | G GTG GGA | AGG AGG | TGA TCC AGT | ATG |
| OB08 | $^{fbaA}$TIR$_6$ *gfp* | *fbaA* | GGA…88bp…ACA | G GTG GGA | AGG GGG | ACA TTC GAC | GTG |
| OB05 | $^{fbaA}$TIR$_{modif2}$ *gfp* | *fbaA* | GGA…88bp…ACA | G GTG GGA | AGG GGG | ACA TTC GAC | ATG |
| OB07 | $^{fbaA}$TIR$_5$ *gfp* | *fbaA* | GGA…88bp…ACA | G GTG GGA | AGG GGG | ACA TTC GAC | TTG |
| OB09 | $^{fbaA}$TIR$_7$ *gfp* | *fbaA* | GGA…88bp…ACA | G GTG GGA | AGG GCG | ACA TTC GAC | GTG |
| OB12 | $^{hs}$TIR$_0$ *gfp* | *hyperspank* | TAA…16bp…ATT | G GTG GGA | AGG AGG | TGA TCC A | ATG |
| OB21 | $^{hs}$TIR$_{short}$ *gfp* | *hyperspank* | TAA…16bp…ATT | G GTG GGA | AGG AGG | AAC TAC T | ATG |
| OB14 | $^{hs}$TIR$_{fbaA}$ *gfp* | *hyperspank* | TAA…16bp…ATT | G GTG GGA | AGG AGG | ACA TTC GAC | ATG |
| OB16 | $^{hs}$TIR$_4$ *gfp* | *hyperspank* | TAA…16bp…ATT | G GTG GGA | AGG AGG | ACA TTC GAC | GTG |
| OB20 | $^{hs}$TIR$_8$ *gfp* | *hyperspank* | TAA…16bp…ATT | G GTG GGA | AGG AGG | GGG TTC GAC | ATG |
| OB13 | $^{hs}$TIR$_{modif1}$ *gfp* | *hyperspank* | TAA…16bp…ATT | G GTG GGA | AGG AGG | TGA TCC AGT | ATG |
| OB18 | $^{hs}$TIR$_6$ *gfp* | *hyperspank* | TAA…16bp…ATT | G GTG GGA | AGG GGG | ACA TTC GAC | GTG |
| OB15 | $^{hs}$TIR$_{modif2}$ *gfp* | *hyperspank* | TAA…16bp…ATT | G GTG GGA | AGG GGG | ACA TTC GAC | ATG |
| OB17 | $^{hs}$TIR$_5$ *gfp* | *hyperspank* | TAA…16bp…ATT | G GTG GGA | AGG GGG | ACA TTC GAC | TTG |
| OB19 | $^{hs}$TIR$_7$ *gfp* | *hyperspank* | TAA…16bp…ATT | G GTG GGA | AGG GCG | ACA TTC GAC | GTG |

*Strains were constructed as indicated in Appendix Table S3.
†The sequence upstream of the pre-sequence depends on the promoter sequence (P$_{fbaA}$ or P$_{hs}$).
‡RBS, ribosome-binding site.
§AR, Accommodation region. The sequence and length of the accommodation region affect ribosomal positioning onto the initiation codon.
‖IC, Initiation codon.
¶Name of the *B. subtilis* strains which contain the GFP under control of the described TIR.

biological replicates, i.e. 24 replicates) grown under identical conditions (Appendix sections 3.1 and 3.2). The ratio of translation efficiencies between strains $^{fbaA}$TIR$_{modif1}$ *gfp* and $^{fbaA}$TIR$_{short}$ *gfp* approximately doubled from 0.3 to 1.5 h$^{-1}$ (Fig 3B), indicating that the number of proteins produced per mRNA and per hour for these two synthetic constructs differently varies with growth rate as a result of only single point mutations in the TIR. We also observed a slight (1.3-fold) increase in the ratio of translation efficiencies between strains $^{fbaA}$TIR$_{modif1}$ *gfp* and $^{fbaA}$TIR$_{fbaA}$ *gfp* from 0.30 to 1.2 h$^{-1}$, which remained invariant thereafter (Fig 3C). Conversely, ratios of translation efficiencies between three other strains remained constant (Fig EV2G and H). Altogether, it indicated that the growth-rate dependency of the translation efficiency depends on the sequence of the translation initiation region.

**Experiment-based estimation of the translation efficiency parameters of the library of synthetic reporter fusions**

Mathematical models of protein production that can handle a transcript-specific, growth rate-dependent translation efficiency (Appendix section 2.5) have previously been proposed (Kremling, 2007; Tadmor & Tlusty, 2008)). What we above defined as the translation efficiency is represented in these models by a Michaelis–Menten rate law, that is $\lambda_i = \frac{K_{1i}[R_{free}]}{K_{2i}+[R_{free}]}$ with two transcript-specific $K_{1i}$ and $K_{2i}$ constants, and where $R_{free}$ (*free* ribosome) abundance corresponds to the fraction of ribosomes ready to initiate translation (Kremling, 2007; Bremer & Dennis, 2008; Klumpp *et al*, 2009). According to a Michaelis–Menten-type translation initiation model, a drop in translation efficiency implies that $R_{free}$ abundance decreases with increasing growth rate. We therefore made use of the model and solved a constrained optimization problem using the GFP expression profiles from a representative subset of our synthetic strains (Table 1) to precisely infer the model parameters and the $R_{free}$ relative abundance with increasing growth rate (Appendix section 3.3). The inferred behaviour of $R_{free}$ abundance exhibited at least a fourfold decrease from slow to fast growth (Fig 4A). A sharp drop occurred between growth rates from 0.25 to 1.2 h$^{-1}$, and $R_{free}$ abundance reached a plateau thereafter. We also obtained a set of $\{K_{1i}, K_{2i}\}$ pairs for each construct that enabled us to accurately fit the corresponding GFP abundances (Fig 4B). Interestingly, the $K_{2i}$ values ranged over 25-fold ($^{fbaA}$TIR$_{modif1}$ *gfp* vs. $^{fbaA}$TIR$_{short}$ *gfp*), which is the result of only a few point mutations

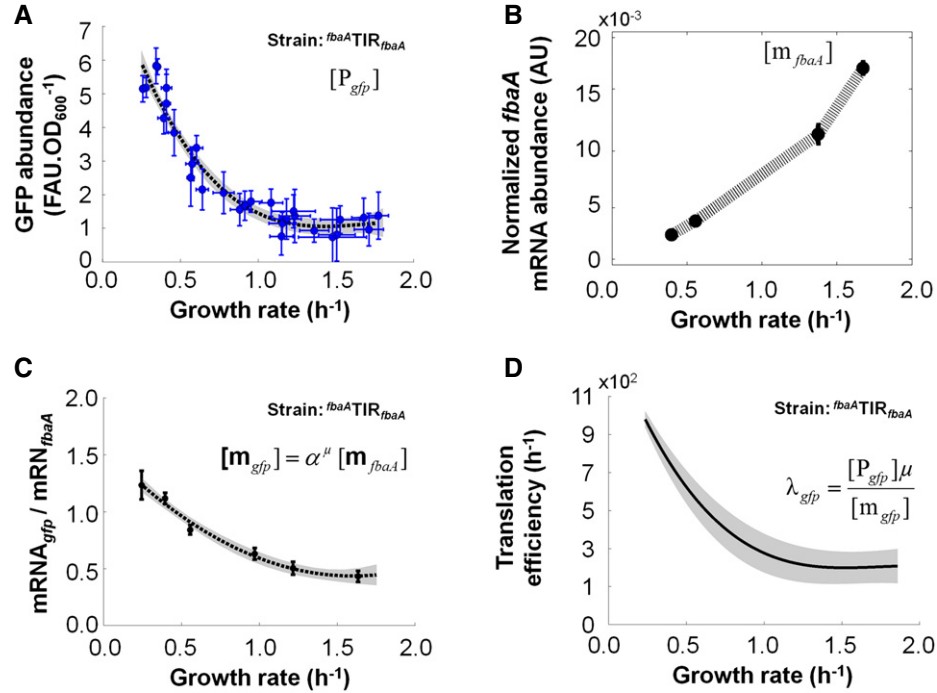

**Figure 2.  Protein abundance and translation efficiency of $^{fbaA}$TIR$_{fbaA}$ drop at fast growth.**

A   Blue dots represent the GFP abundance in different growth conditions (average of 12 technical replicates); the 95% confidence intervals are given as horizontal and vertical bars. Data were fit with a third-order polynomial (dashed line), and the 95% confidence interval of the fit is shown in grey.
B   Normalized *fbaA* transcript abundance (i.e. proportion of *fbaA* transcript within total mRNA × normalized total mRNA abundance from Fig 1D).
C   Relative abundance (measured by qPCR) of the *gfp* mRNA to the endogenous *fbaA* mRNA in strain $^{fbaA}$TIR$_{fbaA}$ as a function of the growth rate. Filled circles are the mean of eight replicates in one medium; the 95% confidence intervals are depicted as bars. Data are fit with a third-order polynomial (dashed line), and the 95% confidence interval of the fit is shown in grey.
D   Deduced translation efficiency (h$^{-1}$) of the $^{fbaA}$TIR$_{fbaA}$ construct as a function of the growth rate. Note that translation efficiency in h$^{-1}$ is a slight abuse of notation since we divided protein abundance (in FAU.OD$_{600}$$^{-1}$) by the normalized transcript abundance (AU, quantity.OD$_{600}$$^{-1}$). The 95% confidence interval is shown in grey. The corresponding datasets can be found in Datasets EV1 and EV2.

within the accommodation region (Table 1). As a consequence, the translation efficiencies showed various profiles as a function of the growth rate (Fig 4C). The translation efficiency of $^{fbaA}$TIR$_{short}$ *gfp* decreased by twofold from slow to fast growth, a behaviour that is clearly explained by a $K_{2i}$ value (of 0.3) in the range of the $R_{free}$ abundance variation (from 0.7 down to 0.2; Fig 4A and B). Eventually, when $K_{2i}$ is very high compared to $R_{free}$ abundance (*e.g.* $^{fbaA}$TIR$_{modif1}$ *gfp*), translation efficiency decreased as much as $R_{free}$ abundance with increasing growth rate. Altogether, this analysis confirmed that translation efficiency can be entirely represented in the form of a Michaelis–Menten-like equation as a function of $R_{free}$ abundance with two TIR-dependent $K_{1i}$ and $K_{2i}$ aggregated constants.

### Exploration of the variety of translation efficiency's growth-rate dependencies in *B. subtilis* using proteomic datasets

We wondered how representative of variation among endogenous *B. subtilis* transcripts are the TIR variants we constructed. We therefore explored the variety of translation efficiency's growth-rate dependencies by combining the above computational approach, and quantitative transcriptome (this work) and proteome datasets obtained from identical growth conditions we recently published in Muntel *et al* (2014) and in Goelzer *et al* (2015). We estimated the

$K_{1i}$ and $K_{2i}$ constants associated with each gene for which the cognate protein has been detected and correctly quantified in at least three growth conditions (*i.e.* 1,002 proteins; Appendix section 3.4 and Dataset EV3). This analysis led us to partition the transcriptome into three classes of transcripts: $R_{free}$-saturated, $R_{free}$-unsaturated and $R_{free}$-undersaturated (Fig 5A). Two-hundred and twenty-three transcripts ($R_{free}$-saturated) exhibited invariant translation efficiencies (*i.e.* independent of the variation in the *free* ribosome abundance: $K_{2i} \ll R_{free}$; Fig EV3A; Dataset EV3). Six hundred and ninety-eight transcripts ($R_{free}$-unsaturated) exhibited translation efficiencies that nonlinearly depend on *free* ribosome abundance (with $K_{2i}$ values in the range of $R_{free}$ abundance). The two transcript-specific $K_{1i}$ and $K_{2i}$ aggregated constants were weakly correlated ($\rho^{Pearson} = 0.60$; $\rho^{Spearman} = 0.56$; Fig 5A; Dataset EV3). Finally, 81 transcripts ($R_{free}$-undersaturated) exhibited translation efficiencies that dropped with increasing growth rate proportionally to the *free* ribosome abundance ($K_{2i} \gg R_{free}$) with $\frac{K_{1i}}{K_{2i}}$ values spanning over several order of magnitude (Fig EV3B; Dataset EV3).

The structural determinants at the ribosome-binding sites are key determinants for the recruitment of the initiation complex onto the mRNA (Milon *et al*, 2012). We therefore wondered whether by higher recruitment of the translation initiation complex, a long 5′UTR may be correlated with low $K_{2i}$ values. In *B. subtilis*,

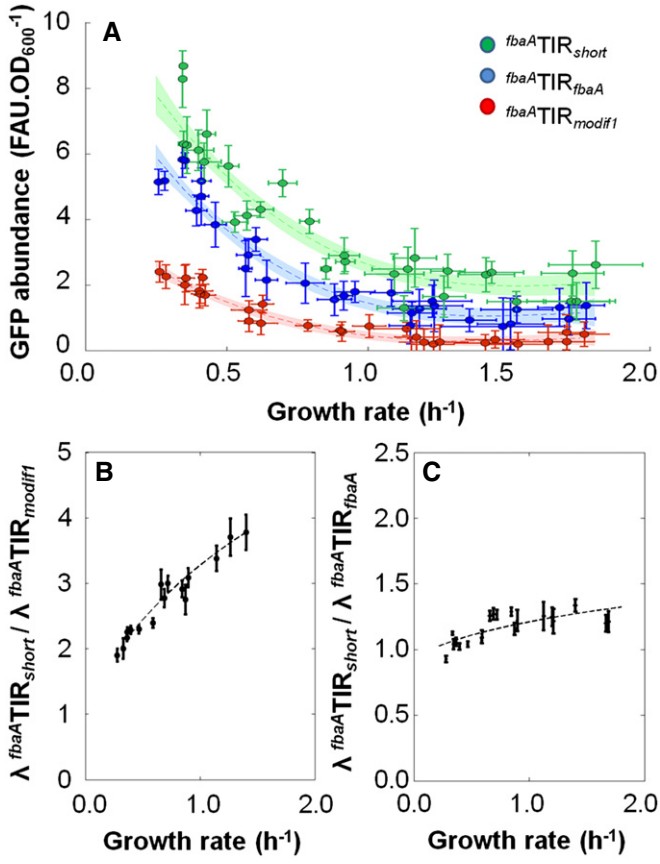

**Figure 3.** Nonlinear translation efficiencies entail differential drops in protein abundances with increasing growth rate.

A    Circles represent the GFP abundance (in fluorescence arbitrary units by optical density units: FAU.OD$_{600}{}^{-1}$) measured during exponential growth for 12 replicates at a given growth rate for constructs $^{fbaA}$TIR$_{fbaA}gfp$, $^{fbaA}$TIR$_{modif1}gfp$ and $^{fbaA}$TIR$_{short}gfp$. The GFP levels were fit with a third-degree polynomial (dashed line); 95% confidence intervals are given in the coloured areas.

B, C    Black dots represent the ratio of the GFP abundance after correction of the different mRNA stability by qPCR (i.e. ratio of translation efficiencies, λ) between two strains ($^{fbaA}$TIR$_{short}gfp$ vs. $^{fbaA}$TIR$_{modif1}gfp$ in panel B; $^{fbaA}$TIR$_{short}gfp$ vs. $^{fbaA}$TIR$_{fbaA}gfp$ in panel C) and the 95% confidence intervals are given as vertical bars (see Appendix section 3.1). The dashed curves are computed using the parameters estimated on Fig 4. The corresponding datasets can be found in Dataset EV2.

one-fourth (157) over barely 600 mapped transcription starts are over 80 nucleotides upstream the coding sequence (Irnov *et al*, 2010). The $K_{2i}$ values of the 71 long 5′UTR messengers identified by our analysis (over the 157 long 5′UTR messengers known so far) were not significantly over-represented in any of the three classes of transcripts (Fig 5A and Dataset EV3). This analysis suggested that long 5′UTR is not the only molecular determinant for the growth rate-dependent regulator-independent global regulation of translation.

We eventually (re)inferred the variation in $R_{free}$ abundance using this extended dataset (Fig EV3C) and obtained, as expected, a relative drop in $R_{free}$ abundance slightly larger than that obtained using the *gfp*-reporter strains from the synthetic library (see Appendix section 3.4). As shown in Fig 5B, the normalized

translation efficiencies for the 1,002 transcripts therefore exhibited various growth-rate dependencies (from invariance to drop). Although we do not exclude the occurrence of a few growth condition-specific post-transcriptional regulation and/or degradation for a subset of proteins, our results demonstrated that *B. subtilis* extensively makes use of the newly identified regulator-independent mode of regulation.

### Structural sensitivity of the global regulation of translation with respect to perturbations

We next wondered whether structurally disturbing the global regulation of translation would affect the translation efficiencies across growth conditions or whether there exists an unknown feedback regulation (most probably acting on *free* ribosome) that would rectify protein production with respect to the rate of growth (Klumpp *et al*, 2009). A straightforward approach to alter the translation process is to use translation inhibitors. When either chloramphenicol or tetracycline was added to the growth medium, Scott *et al* (2010) observed an increase in the total ribosome abundance in *Escherichia coli* to compensate for the inhibition of the elongation phase (Fig 6A). We therefore used tetracycline to disturb the translation process. In the presence of sub-inhibitory concentrations of tetracycline (0.5 and 1 mg l$^{-1}$), the growth rate was reduced and the ratios between the GFP proteins expressed under the control of $^{fbaA}$TIR$_{short}$ *gfp* and $^{fbaA}$TIR$_{modif1}$ *gfp* decreased, as compared to the same growth conditions in the absence of antibiotic (Fig 6B). With 1 mg l$^{-1}$ of tetracycline, the ratio between $^{fbaA}$TIR$_{short}$ *gfp* and $^{fbaA}$TIR$_{modif1}$ *gfp* decreased by 50% in rich medium (CHG) and by 5% in poor medium (M9P; Fig 6B). By contrast, in the presence of 0.5 mg l$^{-1}$ of tetracycline, the ratio only decreased by 25% in rich medium and by 5% in poor medium (Fig 6B). Overall when translation was disturbed in the presence of tetracycline, the ratios between translation efficiencies were significantly altered for all growth rates over 0.4 h$^{-1}$. Altogether, our results indicated that there is no feedback regulation that strictly rectified protein production with respect to the rate of growth, which consequently suggested that the growth-rate dependency of the global regulation of translation resulted from the variation in a molecular entity related to the total ribosome abundance.

### An elementary three-step translation initiation model to explain the different translation efficiency's growth-rate dependencies

In order to provide a rationale for modelling the translation efficiency in the form of a Michaelis–Menten-like equation, we developed a mechanistic model of protein production that decomposes translation initiation into three main molecular steps (Fig 7A, Table 2): i) ribosome binding to the mRNA; ii) accommodation of the ribosome to the start codon; and iii) initiation of translation elongation (Tomsic *et al*, 2000; Ramakrishnan, 2002; Milon *et al*, 2012). Translation initiation is then followed by the translation elongation and termination steps. The underlying molecular assumptions for modelling this process are described in the Appendix (sections 2.1 and 2.2). Due to the succession of reversible and irreversible steps (Fig 7A), the mathematical formalization of this elementary three-step translation initiation process leads to a Michaelis–Menten-type rate law of the translation efficiency ($λ_i$)

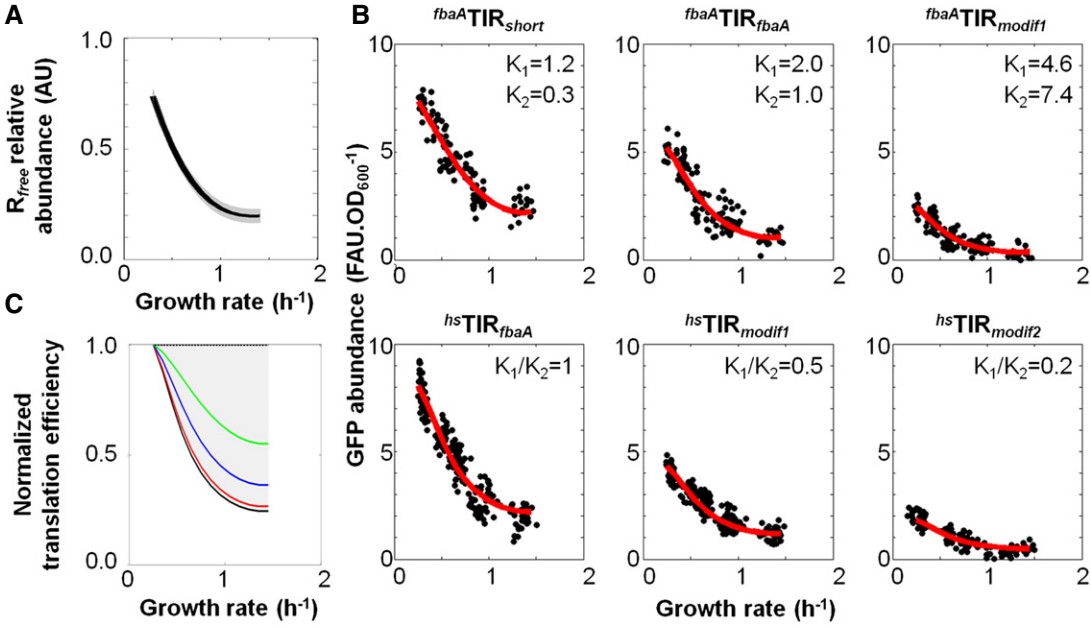

**Figure 4.** $R_{free}$ **abundance drops at fast growth and results in different TIR-specific translation efficiencies.**

A Inference of $R_{free}$ abundance. $R_{free}$ abundance was fit with a third-order polynomial (black line); the 95% confidence interval is given by the grey area (see Appendix section 3.3).

B Set of $\{K_{1i}, K_{2i}\}$ pairs for six synthetic constructs. The three upper plots correspond to synthetic constructs having a translation efficiency as $\lambda_i = \frac{K_{1i}[R_{free}]}{K_{2i} + [R_{free}]}$ and the three lower plots correspond to synthetic constructs having a translation efficiency as $\lambda_i = \frac{K_{1i}}{K_{2i}}[R_{free}]$, with $K_{2i}$ large as compared to $[R_{free}]$. The ratio $K_1^{hsTIRfbaA}/K_2^{hsTIRfbaA}$ was scaled to 1 in order to relatively estimate the entire set of $K_{1i}$ and $K_{2i}$ constants (for detailed explanation, see Appendix section 3.3). Red curves represent model-based fits of GFP abundance quantified in more than 120 cultures for each bacterial strain (black dots).

C Relative translation efficiencies of the six strains from panel (B) ($^{fbaA}$TIR$_{fbaA}gfp$ in blue, $^{fbaA}$TIR$_{modif1}gfp$ in red, $^{fbaA}$TIR$_{short}gfp$ in green and $^{hs}$TIR$_{fbaA}gfp$, $^{hs}$TIR$_{modif1}gfp$ and $^{hs}$TIR$_{modif2}gfp$ in black). The grey area corresponds to the space of possible translation efficiencies, which is bounded by the translation efficiency when $K_{2i}$ is much larger than $R_{free}$ ($^{hs}$TIR$_{fbaA}gfp$, $^{hs}$TIR$_{modif1}gfp$ and $^{hs}$TIR$_{modif2}gfp$) and when $R_{free}$ is much larger than $K_{2i}$ (theoretical values, dashed black line). The corresponding datasets can be found in Dataset EV2.

with a limiting molecular entity, the so-called *free* ribosome abundance ([$R_{free}$], Fig 7B). Following the common consensus used in the study of Bremer & Dennis, 2008; Klumpp *et al*, 2009; Kremling, 2007; Tadmor & Tlusty, 2008; $R_{free}$ corresponds hereafter to the translation initiation complex and is composed of the 30S subunit of the ribosome, the initiation factors IF1, IF2 associated with GTP, and IF3, and the initiator tRNA, fMet-tRNA$^{fMet}$ (30S·mRNA·IFs·GTP·fMet-tRNA$^{fMet}$; Fig 7A). The complete model (Fig EV3D) can be written in the form of a Michaelis–Menten-like equation (Fig 7B) using two integrative, growth rate-independent constants, $K_{1i}$ and $K_{2i}$, which are composed of the primary TIR-dependent constants. It is worth to note that the two integrative constants shared several primary constants (Fig EV3D). Our model encompassed the simplest molecular scenario of the translation process; we nonetheless explored several other scenarios and alternative ribosomal assembly pathways by developing alternative models (Appendix section 2.4). The alternative models gave rise to translation efficiencies in the form of Michaelis–Menten-like equations except that the two aggregated constants ($K_{1i}$ and $K_{2i}$) were constituted of different primary, kinetic parameters and/or that *free* ribosome was constituted of different ribosomal subunits. Altogether, our complete model turned to be a reasonable proxy of the translation process to account for the growth rate-dependent translation efficiencies.

## Interdependence of *free* ribosome, total ribosome and mRNA abundances

The interdependence of *free* ribosome, total ribosome and mRNA abundances may directly set the *free* ribosome abundance across growth conditions due to the growth-rate dependency of both the RNA and ribosome production. We therefore theoretically explored how $R_{free}$ abundance can naturally vary with growth rate by extending our knowledge-based model to the entire proteome (Appendix section 2.3). We demonstrated the formal relationship relating total ribosome ($R_{Tot}$) and mRNA species ($m_i$) with $R_{free}$ (Fig 7C). Although $R_{Tot}$ abundance increases with growth rate (Schaechter *et al*, 1958; Bremer & Dennis, 2008), an analysis of this relationship showed that the two following solutions can trigger a drop in $R_{free}$ abundance (Fig 7D). The first solution consists of a strong increase in the abundances of all or many individual mRNAs with increasing growth rate (as shown in Fig 1D), which at least counterbalances the increase in total ribosome abundance to eventually lower the amount of available ribosomes ($R_{free}$). The second solution consists of a global reorganization of mRNA synthesis (as shown in Fig 1E), during which a class of transcripts (*i.e.* exhibiting low $K_{2i}$ values such as the $R_{free}$-saturated transcripts) is relatively upregulated, while a class of transcripts exhibiting the converse properties is downregulated with increasing growth rate. A likely

**A**

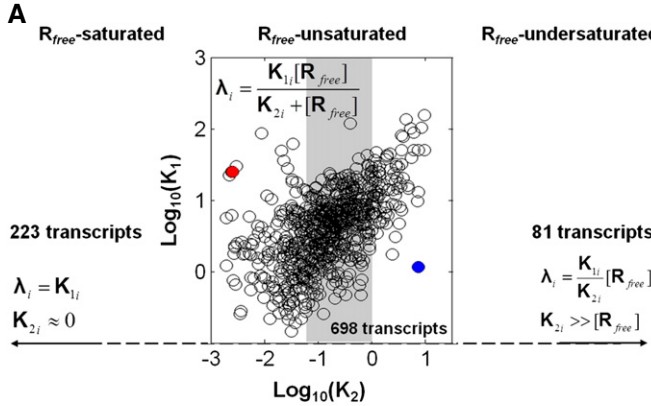

**B**

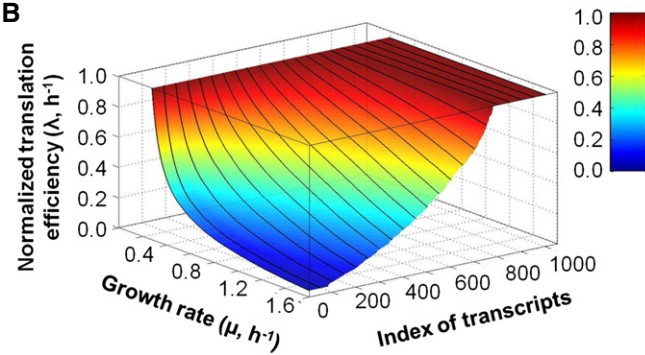

Figure 5.  Genome-wide estimation of {$K_{1i}$, $K_{2i}$} pairs and of over a thousands of growth rate-dependent, gene-specific translation efficiencies in *B. subtilis*.

A    Plot of the values of $K_{1i}$ (in $\log_{10}$) vs. $K_{2i}$ (in $\log_{10}$) for each of the 698 transcripts (*i*), denoted $R_{free}$-unsaturated, for which translation efficiencies were in the form of a Michaelis–Menten-type rate law. For illustration purpose, filled-in circles correspond to the *tufA* (in red) and *sdpC* (in blue) transcripts. The grey area represents the variation in $R_{free}$ (Fig EV3C). 223 transcripts, denoted $R_{free}$-saturated, exhibited a constant translation efficiency (i.e. $K_{2i} \sim 0$, Fig EV3A) and 81 transcripts, denoted $R_{free}$-undersaturated, exhibited a linear relationship between their translation efficiencies and $R_{free}$ (i.e. $K_{2i} \gg R_{free}$, Fig EV3B).

B    Gene-specific, growth rate-dependent translation efficiencies estimated for 1,002 transcripts (*i*, index of transcripts) and normalized to the growth-rate point 0.25 $h^{-1}$ (colour bar and *z*-axis). The plain lines separate transcripts by groups of fifty. The list of the transcripts corresponding to each panel can be found in Dataset EV3.

biological interpretation is that such a reorganization of mRNA synthesis would enhance titration of *free* ribosome at fast growth and eventually lowers the amount of available ribosomes ($R_{free}$). This prompted us to analyse the possible role of ribosomal mRNAs due to the enriched, large proportion of ribosomal proteins in total proteins at fast growth. Our combined statistical and model-based data analyses of genome-wide transcript and protein abundances have identified the $K_{2i}$ values of 32 ribosomal proteins (among about 50 known ribosomal proteins, Dataset EV3). As expected, most of the 32 $K_{2i}$ values were close to zero, which indicated that their translation is barely affected by the rate of growth. Actually, the rate of ribosomal protein synthesis is cross-coordinated with the available amount of rRNA by an autogenous feedback regulation (Nomura's model) in *B. subtilis* (Grundy & Henkin, 1991; Choonee

**A**

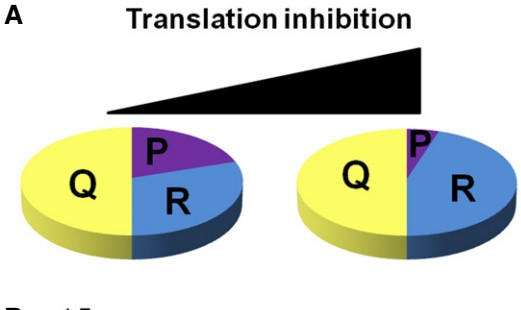

**B**

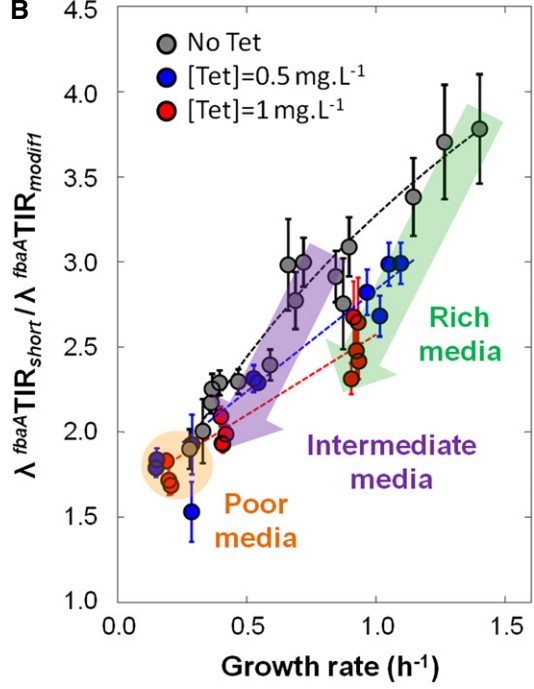

Figure 6.  Challenging the structural sensitivity of the global regulation of translation with respect to perturbations.

A    Inhibition of the elongation phase results in an increase in total ribosome abundance (modified from Scott et al, 2010).

B    Circles represent the ratios of translation efficiencies computed at each growth rate for 12 replicate cultures of the $^{fbaA}TIR_{modif1}$ *gfp* and $^{fbaA}TIR_{short}$ *gfp* strains. The dark grey, blue and red dots correspond to the ratios obtained in the presence of 0, 0.5 and 1 mg ml$^{-1}$ tetracycline, respectively. The green and purple arrows overlay the "ratio vs. μ" variations resulting from the addition of tetracycline in rich (CH, CHG) and intermediate (S, TS) media, respectively. The orange area illustrates the little variation in the ratios in poor media (M9SE and M9P). The corresponding datasets can be found in Dataset EV4. The 95% confidence intervals are given as vertical bars.

et al, 2007) similar to that in *E. coli* (Kaczanowska & Ryden-Aulin, 2007). We then theoretically explored how this feedback regulation interferes with our whole-proteome modelling approach and demonstrated that the enrichment of ribosomal mRNAs with increasing growth rate (Fig 1E) could not alone explain the drop in $R_{free}$ abundance (Appendix section 2.3). As a consequence, if the growth rate-dependent variation in $R_{free}$ abundance only resulted from the trade-off between the titrations of *free* ribosomes with the non-ribosomal and ribosomal mRNAs, several other transcripts contributed to the observed drop in $R_{free}$ abundance. Altogether, our

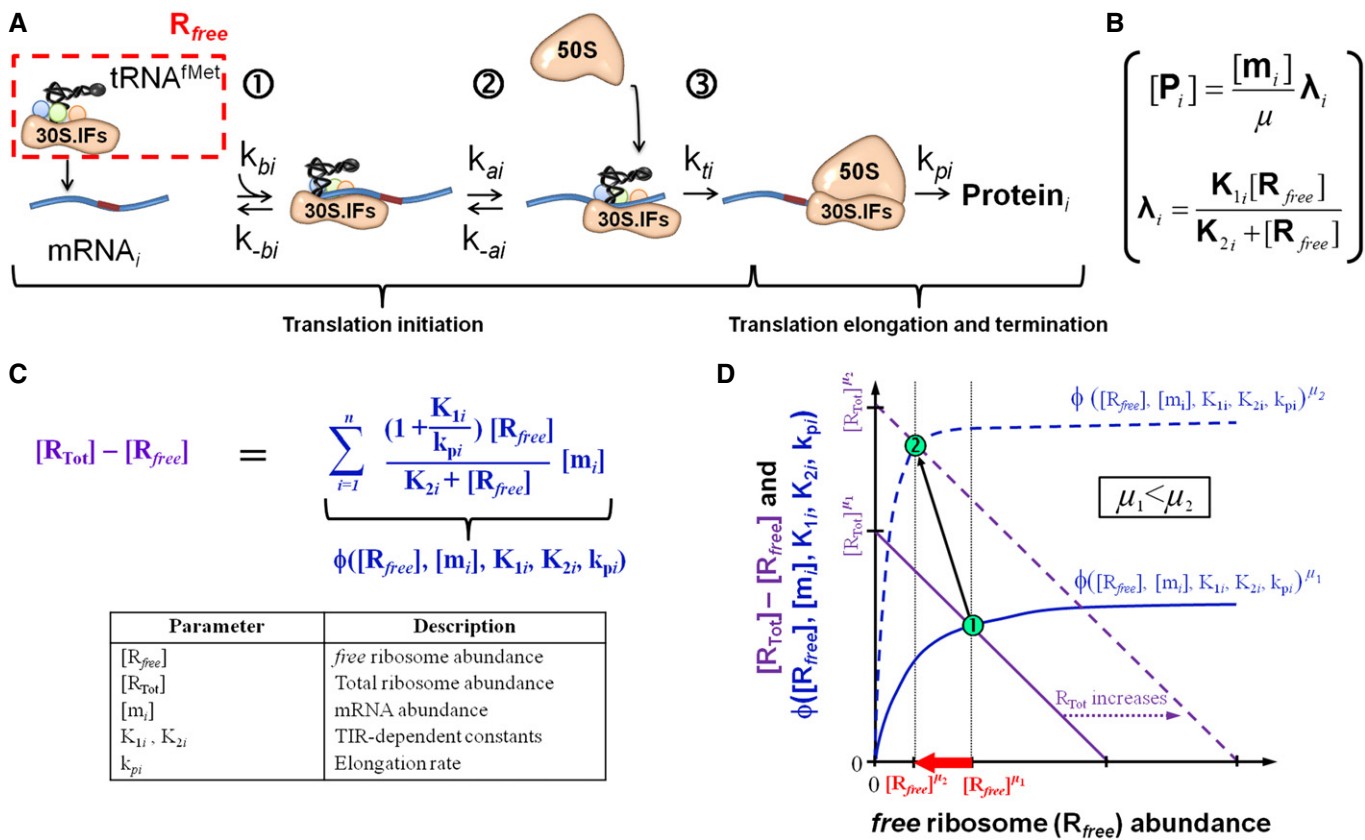

**Figure 7. Exploring the interdependence of *free* ribosome abundance and both mRNA and ribosome abundances using a three-step translation initiation model.**

A   Knowledge-based, three-step translation initiation model (for more information, see Appendix section 2.1). The growth rate-dependent molecular entities and rates of the translation initiation process are the abundance of the translation initiation complex ($R_{free}$), the 50S subunit, the mRNA abundance (mRNA$_i$) and the protein elongation aggregated parameter ($k_{pi}$, Table 2; Bremer & Dennis, 2008). The growth rate-independent parameters are the binding constant of $R_{free}$ onto the mRNA$_i$ ($k_{bi}$), the release constant ($k_{-bi}$) of mRNA$_i$-associated $R_{free}$ ($R_{bi}$), the accommodation constant ($k_{ai}$) of $R_{bi}$ on the start codon, the disaccommodation constant ($k_{-ai}$) of $R_{bi}$-accommodated $R_{ai}$ from the start codon and the constant of the initiation of translation elongation ($k_{ti}$).

B   The elementary three-step translation initiation model gives rise to translation efficiency in the form of a Michaelis–Menten-like equation with the two integrative constants, $K_{1i}$ and $K_{2i}$ (see Fig EV3).

C   Formal relationship relating total ribosome ($R_{Tot}$) and mRNA species ($m_i$) abundances with the *free* ribosome abundance ($R_{free}$; see Appendix section 2.3). The sum of the Michaelis–Menten-like equations (blue) is an increasing function of global and gene-specific variables ($\phi([R_{free}], [m_i], K_{1i}, K_{2i}, k_{pi})$). $n$ indicates the number of different mRNA species.

D   The left and right parts of the equation from panel (C), respectively, in purple and blue, are plotted vs. the *free* ribosome abundance, [$R_{free}$]. The intersection (①) corresponds to the equilibrium in a given growth condition. Following a growth-rate increase from $\mu_1$ to $\mu_2$ ($\mu_2 > \mu_1$), total ribosome abundance increases (from plain to dashed purple line), and the equilibrium (②) is shifted towards a decrease in $R_{free}$ abundance. The global growth rate-dependent transcriptome reorganization (Fig 1E) can directly trigger a drop in $R_{free}$ abundance by increasing titration if a significant fraction of the transcripts from the first class (green) exhibits lower $K_{2i}$ values than the second class (orange).

analysis suggested that the interdependence of *free* ribosome, total ribosome and mRNA abundances is the cornerstone of growth condition-specific cell (re)programming.

# Discussion

Prokaryotes adapt to environmental changes by adjusting their transcriptome in a complex manner mostly *via* the use of DNA-binding regulators that respond to environmental signals or metabolic effectors (Lu *et al*, 2007; Goelzer *et al*, 2008; Buescher *et al*, 2012; Nicolas *et al*, 2012). Yet, the growth rate-dependent variation in abundance of the transcription machinery also strongly influences

gene expression (Klumpp & Hwa, 2008; Gerosa *et al*, 2013). Indeed, total RNA abundance (per mass) is well known to increase with increasing growth rate in both Gram-negative (*E. coli* and *Salmonella typhimurium*) and Gram-positive (*B. subtilis*) bacteria (Schaechter *et al*, 1958; Dauner *et al*, 2001; Bremer & Dennis, 2008). We reassessed total RNA and total mRNA using up-to-date, high-resolution technologies in *B. subtilis*. We confirmed that total RNA abundance increases twofold when the growth rate doubles and showed that total mRNA abundance represents a constant fraction of the total RNA abundance for all tested growth rates (Fig 1C). Altogether, it implies that the total mRNA abundance in *B. subtilis* increases twofold when the growth rate doubles. These results are consistent with a recent investigation reporting that mRNA

**Table 2. Molecular steps of the knowledge-based model of protein production, degradation and dilution.**

| Step | Molecular reaction | Parameters |
|---|---|---|
| ① | Ribosome binding $$R_{free} + m_{fi} \underset{k_{-bi}}{\overset{k_{bi}}{\rightleftharpoons}} R_{bi}$$ | $R_{free}$ = 30S•IF1•IF2•IF3•tRNA$^{fMet}$ (*free* ribosome) $m_{fi}$ = *Free* mRNA coded by the gene $i$ $R_{bi}$ = 30S•IF1•IF2•IF3•tRNA$^{fMet}$•m$_i$ (active ribosome) $k_{bi}$ = rate of *free* ribosome binding $k_{-bi}$ = rate of *free* ribosome release |
| ② | Ribosome accommodation $$R_{bi} \underset{k_{-ai}}{\overset{k_{ai}}{\rightleftharpoons}} R_{ai}$$ | $R_{ai}$ = 30S•IF1•IF2•IF3•tRNA$^{fMet}$•m$_i$ (pre-initiating ribosome) $k_{ai}$ = rate of ribosome positioning onto the start codon $k_{-ai}$ = rate of ribosome release from the start codon |
| ③ | Initiation of translation elongation $$R_{ai} + 50\,S_{free} \overset{k_{ti}}{\longrightarrow} R_{ti}$$ | $R_{ti}$ = 50S•30S•IF1•IF2•IF3•tRNA$^{fMet}$•m$_i$ (initiating ribosome) $50S_{free}$ = *free* 50S subunit of the ribosome $k_{ti}$ = rate of initiation of translation elongation |
| | Completion $$R_{ti} \overset{k_{pi}}{\longrightarrow} P_i$$ | $P_i$ = protein coded by the gene $i$ $k_{pi}$ = protein elongation aggregated parameter (equal to the inverse the population averaged time of protein elongation) |
| | Degradation $$P_i \overset{\gamma_i}{\longrightarrow} \varnothing$$ | $\gamma_i$ = rate of protein degradation |
| | Dilution $$P_i \overset{\mu}{\longrightarrow} \varnothing$$ | $\mu$ = growth rate |

abundance in *E. coli* at least doubles when growth rate increases from 0.11 to 0.49 h$^{-1}$ (Valgepea *et al*, 2013).

**Translation efficiency based on Michaelis–Menten kinetics: the what, why and how**

In Gram-positive and Gram-negative bacteria, the drop in abundance of constitutively expressed proteins is comparable ((Bremer & Dennis, 2008; Klumpp *et al*, 2009; Scott *et al*, 2010) and this work). In this study, we demonstrated that this drop in *B. subtilis* is not only due to the dilution but also to an unexpected decrease in translation efficiency (Fig 2). We also observed that the translation efficiencies of two proteins can exhibit different growth-rate dependencies (Fig 3B and C). To provide a rationale to the observed translation efficiency's growth-rate dependencies (Appendix Fig S1) across normal growth conditions or upon antibiotic addition (Figs 3 and 6), we developed and analysed an elementary knowledge-based model of translation in which translation initiation is decomposed into three main steps. Such formalization ensued from straightforward, reasonable biological assumptions and took into account the commonly accepted scenario of ribosome assembly and translation initiation *in vivo*. However, several others scenarios and pathways of ribosome assembly have recently been characterized *in vitro* with *E. coli* cell-free extracts (Tsai *et al*, 2012). We thoroughly explored these

scenarios and the alternative ribosomal assembly pathways by developing alternative models (Appendix section 2.4). The alternative models also gave rise to translation efficiencies in the form of Michaelis–Menten-like equations except that the two aggregated constants ($K_{1i}$ and $K_{2i}$) were constituted of different primary, kinetic parameters. Altogether, it means that modelling the translation efficiency in the form of a Michaelis–Menten-like equation is relevant to investigate bacterial translation. Interestingly, an early study from the nineties in *E. coli* also tends to support the existence of differential protein productions (Jacques *et al*, 1992), which in turn supposes a variation in $R_{free}$ abundance in Gram-negative bacteria. In order to draw firm conclusions, dedicated experiments should now be performed with *E. coli* to confirm in Gram-negative bacteria that translation efficiencies can exhibit different growth-rate dependencies, especially by altering the TIR-related $K_{1i}$ and $K_{2i}$ constants and $R_{free}$ abundance.

Our whole-proteome modelling approach suggested that the global regulation of translation mediated by the growth rate-dependent variation in $R_{free}$ abundance may only result from the interdependence of the *free* ribosome, total ribosome and mRNA abundances across the growth conditions. Because of the strong increase in total ribosome abundance with increasing growth rate, the increase in ribosomal mRNA abundance could not trigger alone the observed drop in *free* ribosome abundance. This suggested that the growth rate-dependent reorganization of the expression of each transcript in the cell is specifically hard-coded on the genome to set the proper amount of *free* ribosomes for a given growth condition. As a consequence, there might not exist any specific feedback regulation controlling the abundance of *free* ribosome and therefore the translation efficiency's growth-rate dependencies of each transcript in the cell. As a corollary, perturbation of the growth rate-dependent variation in the *free* ribosome abundance is expected to alter the gene-specific translation efficiency's growth-rate dependencies. In this study, we demonstrated that the gene-specific translation efficiency's growth-rate dependencies were sensitive to the presence in the growth media of sub-inhibitory concentrations of tetracycline. It is tempting to postulate that tetracycline modified the growth rate-dependent drop in the *free* ribosome abundance, which in turn altered the gene-specific translation efficiency's growth-rate dependencies. We therefore inferred the growth rate-dependent evolution of the *free* ribosome abundance across growth conditions in the presence of 1 mg l$^{-1}$ of tetracycline, which now exhibited a maximum twofold decrease from slow to fast growth (Fig EV3E). However, we cannot exclude that tetracycline affected translation initiation in a way that altered the two $K_{1i}$ and $K_{2i}$ growth rate-independent aggregated constants. Nevertheless, in the presence of tetracycline cells were unable to set right the production of proteins with respect to the rate of growth. Altogether, it suggests that there is no strong feedback control on the abundance of *free* ribosome across growth conditions.

**The space of translation efficiencies allows *B. subtilis* cells to adjust single protein abundances across growth conditions**

Our combined statistical and model-based data analyses of genome-wide transcript and protein abundances uncovered a large range (several orders of magnitude) of $K_{2i}$ aggregated constants and revealed that the drop in *free* ribosome abundance altered

genome-wide the translation efficiency's growth-rate dependencies. According to the model, the translation efficiency can be constant at any growth rates if the $K_{2i}$ value is very small as compared to that of $R_{free}$. If such is the case, the gene-specific translation efficiencies simplify into the $K_{1i}$ values and the current model of Hwa and colleagues (Scott *et al*, 2010) perfectly describes the process of protein production. Our results showed that it was indeed the case for about one-fourth of the transcripts (*i.e.* 223 genes) for which we have characterized the $\{K_{1i}, K_{2i}\}$ pairs. In addition, several other transcripts (about 100 among the 698 transcripts from Fig 5A) are only barely affected by the drop in *free* ribosome abundance, so that the resulting dilution-corrected protein abundances may also be perceived as invariant. Remarkably, the 1,002 transcript-related translation efficiencies fully filled in the space predicted by the three-step translation initiation model between $K_{1i}$ and $\frac{K_{1i}[R_{free}]}{K_{2i}}$ (Fig 5B vs. Fig 4C). Till now, translation initiation regions were believed to be shaped by evolution in order to optimize the trade-off between the level of expression and the resulting gene expression noise. The gene expression noise is indeed important for the highly translated transcripts (Ozbudak *et al*, 2002; Ferguson *et al*, 2012). However and in view of our results, TIRs must also be regarded as targets for selective evolution towards optimized cellular networks, in particular by efficient adjustment of the growth-rate

dependencies. An open issue is whether evolutionary constraints provided more incentives to set protein abundances across growth conditions (see Appendix section 3.5) or the resulting noise, and whether independently modulating the $K_{1i}$ and $K_{2i}$ variables for a given protein abundance will generate different levels of noise.

### Transcriptional and translational growth rate-dependent global regulations optimize cellular fitness without dedicated regulators

The growth rate-dependent regulation (also referred to as global regulation) of bacterial translation provides cells with a remarkable toolbox to tweak protein production. However, translation efficiency is either a constant or a decreasing function of growth rate. By contrast, the global regulation of transcription *via* the transcription efficiency is an increasing function of growth rate (Klumpp & Hwa, 2008; Gerosa *et al*, 2013). When combined, the global regulations of transcription and translation may provide cells with a much larger range of protein expression in the absence of dedicated regulators and may allow prokaryotes to fine-tune the abundance of each protein as a function of the growth rate even in the absence of dedicated regulators (Figs 8, EV4 and EV5). We can therefore question why do cells need dedicated regulators? By

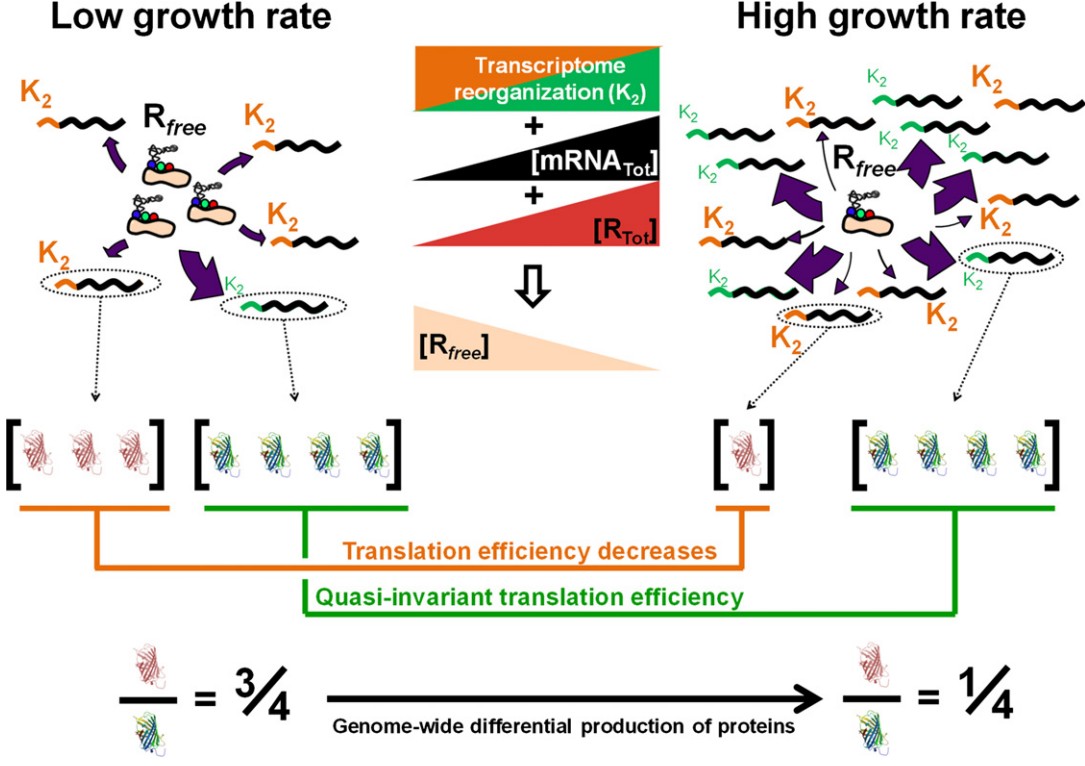

**Figure 8.  Schematic representation of the interdependence of *free* ribosome abundance and both mRNA and ribosome abundances.**
The $K_2$-related global reorganization of mRNA expression leads to a drop in $R_{free}$ with increasing growth rate, which in turn promotes differential protein production. From low to high growth rates, the total mRNA and total ribosomal abundances as well as the protein elongation aggregated parameter ($k_{pi}$, Table 2; Bremer & Dennis, 2008) increase, while the *free* ribosome abundance drops. Titration of $R_{free}$ is weaker at low than at fast growth because transcripts exhibiting strong $K_{1i}$ and low $K_{2i}$ (green TIR) are relatively upregulated and those exhibiting converse properties (orange TIR) downregulated with increasing growth rate. The ribosome density along the mRNA is the combined outcome of the translation efficiency and the elongation rate. The "red" and "green" proteins are differentially produced with increasing growth rate due to the growth rate-dependent translation efficiency. See also a more detailed representation in Fig EV5.

responding to specific signals, regulators allow bringing discontinuity in protein expression across growth conditions, as opposed to the nonlinear but continuous transcription and translation efficiencies functions. Evolutionary theories suggest that protein expression levels maximize fitness (Dekel & Alon, 2005; Molenaar *et al*, 2009; Goelzer *et al*, 2011; Goelzer & Fromion, 2011). In particular, the growth rate-dependent regulations may contribute to cost minimization by providing reasonable solutions to the cost–benefit optimization of the cellular and metabolic processes that must be active under all growth conditions and consequently to fitness increase across growth conditions (Goelzer *et al*, 2015).

We revealed that the variations in the *free* ribosome abundance can globally contribute to proper cellular (re)programming and have evidenced a unique, hard-coded, growth rate-dependent mode of translation regulation, which manages genome-wide gene expression in addition to specific post-transcriptional and translational regulations. Our findings on the global regulation of translation, together with previous findings on the global regulation of transcription (Klumpp & Hwa, 2008; Gerosa *et al*, 2013), will open pioneering opportunities for the differential modulation of complex synthetic circuits. By coupling high-throughput, precise genome editing technologies, we can now envision a rational, *in silico* design and experimental (re)programming of a high performing cell factory genomically streamlined to optimize growth and production properties at a given growth rate.

# Materials and Methods

## Biological materials

*Escherichia coli* TG1 was used for plasmid constructions and transformations using standard techniques (Sambrook *et al*, 1989). A tryptophan prototrophic *B. subtilis* 168 (BSB168) strain (Botella *et al*, 2010) was transformed using standard procedures (Anagnostopoulos & Spizizen, 1961). The reference OB1 strain, containing the $P_{hs}$ (IPTG-inducible *hyperspank* promoter), derived from BSB168 by double crossover insertion in the *amyE* locus of the pOB1 plasmid. To generate the Cm$^R$ pOB1 plasmid, the pDR111 plasmid (kind gift of David Rudner), which carries the $P_{hs}$ and *lacI* gene between two arms of the *amyE* gene, was digested by *Sac*II/*Eco*RI and the 3,083-bp fragment was sub-cloned into the *Sac*II/*Eco*RI sites of pDG1661 (Guerout-Fleury *et al*, 1996). To generate vectors that contained the *gfpmut3* gene downstream of a given translation initiation region (TIR), we amplified by inverse PCR the entire pBaSysBioII plasmid (Botella *et al*, 2010) using primers introducing point mutations in the original TIR but conserving the ligation-independent cloning site (LIC). For LIC, the vector-related PCR products were gel-purified, treated with T4 DNA polymerase and 2.5 mM dATP, and resulting fragments phosphorylated with T4 polynucleotide kinase. Promoter regions (400 bp) of *fbaA* ($P_{fbaA}$) and *hyperspank* ($P_{hs}$) were generated by PCR from OB1 genomic DNA using the appropriate primers listed in Appendix Table S2. Then, 0.2 pmol of each PCR amplification was incubated with 2.5 mM dTTP and T4 DNA polymerase. A mix of 5 ng of prepared vector-related fragment and 15 ng of promoter-related PCR amplification was used to transform *E. coli*. The resulting plasmids were extracted from *E. coli*, used to transform OB1 strain by single crossover either at the *fbaA* locus or at the *hyperspank* locus (within *amyE*), leading to the strains listed in Table 1.

## Growth conditions

LB was used to grow *E. coli* and *B. subtilis*. The eight growth media used for *B. subtilis* strain to reach various growth rates were modified from previously described growth media (Harwood & Cutting, 1990; Partridge & Errington, 1993; Sharpe *et al*, 1998; Kleijn *et al*, 2010; Chubukov *et al*, 2013). The composition of the different media is described in Appendix (section 1.1). When required, media were supplemented with antibiotics at the following abundances for *B. subtilis*/*E.coli*: 100 μg ml$^{-1}$ of ampicillin (only for *E. coli*), 200/100 μg ml$^{-1}$ of spectinomycin or 20/5 μg ml$^{-1}$ of chloramphenicol. The related bacterial growth rates were 0.25 h$^{-1}$ in M9P (M9 pyruvate), 0.40 h$^{-1}$ in M9SE (M9 succinate/glutamate), 0.60 h$^{-1}$ in S, 0.75 h$^{-1}$ in M9G (M9 glucose), 0.80 h$^{-1}$ in M9M (M9 malate), 0.90 h$^{-1}$ in TS, 1.20 h$^{-1}$ in CH and 1.70 h$^{-1}$ in CHG. To alter ribosome abundance, the M9P, M9SE, S, TS, CH and CHG media were supplemented with 0.5 or 1 mg l$^{-1}$ of tetracycline.

## Quantification of RNA molecular species

RNA extraction was modified from Nicolas *et al* (2012) (Appendix section 1.4). Total RNA quantification was performed using a Nanodrop ND-1000 spectrophotometer (Thermo Scientific). RNA quality and rRNA quantification were assessed with an Agilent 2100 Bioanalyzer (Agilent Technologies). In order to scale the gene-level intensities from each microarray and to determine the total mRNA fraction out of the total RNA pool, an equal amount of *in vitro*-synthesized transcripts (One Color RNA Spike-In Kit, Agilent Technologies) was added to 10 μg of each total RNA sample. Synthesis, one-colour hybridization of fluorescently labelled cDNA to Agilent custom microarrays and data processing are described in Appendix (section 1.4). Experimental procedure for real-time quantitative PCR is depicted in Appendix (section 1.5).

## Protein abundance determination by Live Cell Array

For live cell array experiments, a single colony of *B. subtilis* was grown in a well of a 96-well microtitre plates (Cellstar$^®$, Greiner bio-one) with Luria–Bertani (LB) medium until an OD$_{600}$ of 0.4–0.5. For precultures in the medium of interest, LB-grown cells were diluted 400-fold into 96-well microtitre plates and incubated overnight under constant shaking at 37°C until OD$_{600}$ reached 0.3. The cultures, with a dilution that yielded exponentially growing cultures next morning, were diluted in 100 μl of the same medium to an OD$_{600}$ of 0.001 into 96-well microtitre plates and incubated at 37°C with constant shaking in a Synergy™ 2 multimode microplate reader (BioTek) for at least 20 hours. OD$_{600}$ and fluorescence (excitation: 485/20 nm, emission: 528/20 nm) were measured at an interval of 7 minutes. OD$_{900}$ and OD$_{977}$ were measured once at the beginning of the experiment in order to correct the optical path length to 1 cm using the following equation: (OD$_{977}$-OD$_{900}$)/0.18. Data were extracted and processed as previously described (Aichaoui *et al*, 2012; Botella *et al*, 2010; Buescher *et al*, 2012; see Appendix sections 1.6 and 1.7).

## Data availability

Data generated in this work are provided in Dataset EV files. The microarray data have been made publically available in Gene

Expression Omnibus (GEO) database with the accession number GSE78108. Proteomic datasets used in this study are available in Goelzer *et al* (2015).

**Expanded View** for this article is available online.

## Acknowledgements

We are grateful to Terence Hwa for his valuable advice and to Ivan Mijakovic and Mahendra Mariadassou for discussions during the course of this work. We would like to specially thank Rut Carballido-López, Marc Dreyfus, Marie-Agnès Petit, Uwe Sauer, Mathias Springer and Massimo Vergassola for critical reading of the manuscript. We would also like to thank the reviewers for improving the quality and readiness of the manuscript. This work was supported by an INRA doctoral grant (CJS to O.B.) and by the European Commission 7[th] Framework project BaSynthec (FP7-244093).

## Author contributions

OB constructed the library of plasmids and strains, collected data, performed qPCR and transcriptomic studies, mathematical modelling and analysed data; AG was involved in mathematical modelling; MS and UM performed RNA hybridization; MC collected data; SA was involved in study design and discussion of the results; and MJ and VF designed the study, analysed data and wrote the paper.

## Conflict of interest

The authors declare that they have no conflict of interest.

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
