## [Review Process File · Molecular Systems Biology]

Translation elicits a growth-rate-dependent, genome-wide, differential protein production in *Bacillus subtilis*

Olivier Borkowski, Anne Goelzer, Marc Schaffer, Magali Calabre, Ulrike Mäder, Stéphane Aymerich, Matthieu Jules and Vincent Fromion

Corresponding author: Matthieu Jules, Micalis Institute, INRA and Vincent Fromion, MaIAGE, INRA

Review timeline:

Submission date:	01 October 2015
Editorial Decision:	15 November 2015
Revision received:	18 February 2016
Editorial Decision:	15 April 2016
Revision received:	18 April 2016
Accepted:	20 April 2016

Transaction Report:

1st Editorial Decision

15 November 2015

Thank you again for submitting your work to Molecular Systems Biology. We have now heard back from the three referees who agreed to evaluate your manuscript. Overall, the reviewers appreciate that you address a timely topic. However, they raise a series of concerns, which should be carefully addressed in a revision of the manuscript.

Without repeating all the points listed below, as you will see, reviewer #3 raises two rather fundamental issues, which are related to the main conclusions of the manuscript. While at this point it is somewhat unclear whether these issues are addressable, we think that you should be given the opportunity to submit a major revision. The two issues, which refer to the findings presented in Figures 6 and 7, are the following:

- Further analyses are required to support the proposed genome-wide relationship between translation efficiency and Rfree abundance.
- The purpose and interpretation of the experiments in the presence of tetracycline is not so clear to reviewer #3 and to us (after re-reading the paper and the reviewers' comments). It also remains unclear whether the conclusions derived from this experiments are well supported. Since these issues are related to key conclusions of the study, they should be convincingly addressed.

Moreover, in line with the comments of reviewers #1 and #3, we would ask you to make sure that the manuscript is carefully re-written in order to make the main findings accessible to a broad audience. As you will see below, reviewer #1 mentions that s/he had a hard time understanding the study, and as such s/he mostly made relatively minor comments.

 REFEREE COMMENTS

Reviewer #1:

I believe that the findings that the authors made are of importance and this could at some point become a contribution for MSB. However, at its current state, the manuscript is not yet suitable for MSB. The text and the analyses are so convoluted that it is very difficult for a reader to understand and appreciate the work. Even a referee that really made an attempt and spent numerous hours on reviewing could not really appreciate (and evaluate!) the work.

I strongly advise the authors to rewrite their manuscript, to adhere to the good practice of presenting ONE idea in ONE paragraph (not having paragraphs that go over more than one page) and to adopt a strict and logical structure.

Below, please find a list with points that can serve as an indication for the things that prevented me from evaluating the work in depth. In my opinion the authors do not only need to address these points, but they need to completely rewrite the manuscript so that it can be evaluated.

Next to this point, the authors also need to check one critical assumption that they made: p.9, supplementary text 2.1: The authors assume that dilution due to growth is much higher than protein degradation. While this might be correct for the high growth rates where this was probably tested, the question is whether this is also the case at lower growth rates. Given the importance of this assumption for the overall conclusion of this work, I feel that the authors need to experimentally test this assumption.

I cannot exclude that more critical issues will be identified once a re-written version can be evaluated. I recommend rejecting this work for publication in MSB. If the authors can provide a completely reworked version that then can be evaluated, MSB should allow for a resubmission.

First result paragraph; long, weird logic ... mix of method description, speculation, results, ... make a clear structure. It took me very long to get it. In combination with Units (relative, normalized), inconsistent use of terms make things very confusing; in Fig 1D: on the two y-axes two different units.

From Fig S1A it seems that the authors have measured at different growth rates than the data shown in Fig 1A seems to indicate? Were these the same experiments? I would at least expect in Fig 1A to also see horizontal error bars.

Fig S1B: The description on y-axis is confusing. If you say "rRNA distribution (%)" I expect values to sum up to 100%.

p.5: The authors assume on page 5 around lines 9ff that the proportion of tRNA on total RNA is as constant over growth rates as it is in *E. coli*. I find this a critical assumption, particularly because they show in Fig 1A that the total RNA abundance in *B. subtilis* behaves completely different across growth rates as in *E. coli*. I am particularly concerned that the authors state in line 14/15 that the mRNA abundance will increase faster in *Bsub* than in *Ecoli*. I would simply remove this whole speculation (i.e. lines 9 to 16) and directly go and measure it.

Description of the data shown in Fig 1F: The text is completely decoupled from what is shown in the figure; genes are mentioned (*dhaS* that are not shown in the figure; how can I see 2x more enrichment of *fbaA*, "at the boundary of the two classes"?)

Comments on Figure 1 (note similar things would apply to other figures as well): Some of the data is normalized to OD. And such normalized data is compared with respectively normalized data from *E. coli*. I find this problematic as ODs can be measured at different wavelength, even when measured at the same wavelength different spectrophotometers can give different readings, and finally growth-rate dependent cell morphology differences can also give different OD readings. Thus, in this quantitative work, I feel the authors should use more accurate normalizations.

Fig. 1A: - add "total" to the y-axis

Fig. 1D: add "total" in front of "RNA" on the y-axis, what does "uOD" mean (i.e. the "u")? Why are there different units on the 2 y-axes. How can I then compare the different values?

Fig. 1C: Description of y-axis unclear

Fig. 1F: Not clear from the y-axis description what the data is normalized to?

Fig. 1E: μ_{ue2} is indicated to be 1.6, while in the text it is stated that the growth rates measured were 1.7. Furthermore, the shown equation is never mentioned and it is not clear to me what it is supposed to show me.

Note that the figures in Fig 2 carry different units for the OD, indicating wavelength and U.OD instead of uOD.

P.7, line 1: start with defining translation efficiency and motivate it

p 7. Line 10 "combining total mRNA quantification and qPCR-corrected transcriptome data I am lost. I am confused: Because in this sentence it seems that they determine *gfpmut3* transcript abundance, but a few lines earlier they already mentioned that they did it and even showed a comparison in Fig 2B....

p.7 line 12: They then "deduced the translation efficiency". The authors have neither told the reader how they have defined translation efficiency nor did they tell the reader how they calculated. Also because there are no units on the y-axis of Fig 2C, I cannot deduce it myself. (It is only hidden in the caption of supplementary Figure 2).

p.7. lines 13-17, the mentioned data is not shown; further the authors don't provide any motivation for this experiment

Confusing: Is ribosomal binding sites the same as translational initiation regions?

P8. Line 9: it sounds the authors have generated new strains, but in fact the same strains were mentioned a page earlier

P8. Line 11: The way this sentence is phrased it sounds like the authors did something, but in fact they refer to a reference. This is incorrect use of a citation.

P8. Line 18-20: Here, the authors mention that they do a correction and then refer to the panel of supplementary Fig 3A-D. Here data is shown, but in no way explained how the correction was done. The reader is left with detective and inference work to find out what the authors meant and did. Note, at this point of the manuscript I am already 2 hours busy in doing this detective work, but I still do not really understand it.

p.8 lines 21-25, p.9. lines 1-14: First a computational analysis is mentioned in the paragraph, then they jump to experimental data and draw some new conclusions from this data. Where is the logic? Why this jumping??? The computational analysis mentioned refers to obtaining statistical confidence, but I see no where any statistical significance measure - neither in the text nor in the figure Fig 3B.

p.9 line 19: The authors mention 3 steps and refer to Fig 4A, where 4 (!) steps are shown.

Furthermore, the figure also shows the 50S unit, which is in turn not mentioned in the text. All of this is confusing.

Caption of Fig 4A, they mention the rate of elongation (k_{pi}). First, k_{pi} is a parameter and not a rate. Second, what is the mechanistic basis of its growth rate dependence?

p.11. Lines 3-4: They conclude that modeling translation as an elementary three step initiation process can reproduce the data. However, on the page earlier they made a statement that a reversibility is crucial to achieve this. What is now correct?

p.11 Lines 7-8: Where can I see this? How can I understand this?

p.11, lines 10-11. Not clear how this was done. What was optimized?

p.11, lines 19: The authors talk about sensitivities and refer to Fig 5C but here I cannot see sensitivities. After some thinking, I think the authors mean that variations in parameter values can change the growth-rate dependencies of the translational efficiency.... But why the heck don't the authors simply say this and spare the reader some unnecessary thinking...?

p.12, lines 13-14: It is completely unclear how I should derive the conclusion mentioned here from the Fig. 6A.

p.12, This paragraph is again very convoluted and the authors jump between different things: First, they mention that they wanted to estimate the parameters; then they make a statement on nonlinearity of certain transcripts' translation efficiency, and in the next sentence they make a statement that the two K parameters are correlated, and finally in the next sentence they make a yet unrelated statement on two specific transcripts. Either I don't get it at all, or this is very confusing.

p.12/13. After the authors have mentioned Fig. 6A in their text, they jump to Fig.8 leaving out the rest of the figures in Fig 6 and the Fig 7. This is very unprofessional and very confusing.

p.13. lines 10-17: I am completely lost. Are the authors again jumping?

p.13, line 24: What is a graphical analysis? What was done here? What do they mean by an "event"?

p.14, line 7: what is a global reorganization of mRNA synthesis?

p.14: What is described in this section would rather belong to the discussion.

p. 14, lines 16-17. Where can this be seen? What is "significant"?

p.14, lines 17-20. Where can this be seen?

Reviewer #2:

This manuscript addresses growth rate dependencies of protein and mRNA concentrations as well as their ratio, the translation efficiencies in *B. subtilis*. It reports the unexpected result that translation of individual mRNAs changes differentially as a function of growth rate, which is interpreted as reflecting a changing concentration of free ribosomes and mRNAs with ribosome binding sites that have different parameters in a Michaelis-Menten type model. These results are obtained by a nice combination of transcriptome and proteome wide data, analysis of gfp constructs with mutated ribosome binding sites and mathematical modeling. In my opinion, the manuscript is very well suited for publication in *Mol Sys Biol*.

I have reviewed a previous version of this manuscript for another journal. Since then the manuscript has been substantially modified and extended. In particular, the concerns I had with the introduction and the discussion of the previous version have been resolved by the revision.

However, I still have two issues with this manuscript.

It is not clear to me how the elongation of the peptide chain enters the analysis of the proteome data. In the model, elongation is included in the rate $k_{p,i}$, which should depend both on the speed of elongation (and thus on codon usage, tRNA concentrations etc) and on the length of the transcript. The authors state in the supporting information that this parameter does not affect the Michaelis-Menten behavior of the translation rate and I agree with that statement as long as R_{free} is considered as known. However, for the global model, where R_{free} is also unknown, the values of the $k_{p,i}$ will affect R_{free} . Thus, I think $k_{p,i}$ values should somewhere be obtained from the proteome analysis and I do not see where. In my opinion, this point needs an explanation. As a consequence, some properties attributed to ribosome binding sites could reflect sequences properties further downstream.

A minor issue is whether the increase of ribosome concentration upon growth in tetracycline has been tested. I think this is a necessary control, because the paper of Scott et al. reported that effect in *E. coli* and the control of ribosome synthesis is (partly) different in *B. subtilis* than in *E. coli*.

Reviewer #3:

The authors report an interesting study on the dependency of translation (initiation) efficiency/rate - the translation rate per mRNA - on growth rate in *Bacillus subtilis*. They argue that:

1. the translation-efficiency/growth-rate dependency in *B. subtilis* is transcript specific and arises via a non-regulated, passive phenomenon,
2. the mechanistic origin of this dependency derives from the reduction of the free ribosome concentration with growth rate, which affects translation initiation rate,
3. the growth-rate dependence of the translation efficiency of transcripts varies between transcripts because their translation-efficiencies have a different dependence on the free ribosome concentration,
4. The free-ribosome-concentration dependence of the translation efficiency of different transcripts varies when their translation-initiation-region sequences vary, giving rise to different translation-initiation kinetics and a different dependency of the initiation rate with the free ribosome concentration.

The authors use experiments and modelling to illustrate and substantiate those claims.

I outline my general understanding of the theoretical concepts and experiments underlying this work one by one, I do this because the authors are sometimes a bit sloppy with their use of quantitative concepts, what their units are, and which modelling assumptions are made. For instance, the key concept of this work is the "translation efficiency", it is however used in two consecutive sentences in different ways: on page 3, line 23 and 24 the translation efficiency is defined as the number of proteins produced per mRNA, whereas in the next sentence, on page 4, it is defined as a rate, namely as the number of proteins produced per mRNA per unit time. I take the latter to be the correct definition. I propose that the authors define those concepts unambiguously with units and symbols, perhaps in a table or box. When I do this my understanding of the paper is as follows. At

balanced growth, the concentration of proteins, P , is fixed such that its relation with its cognate mRNA concentration, M , the growth rate, g , the protein degradation rate, k_p , and the translation rate per mRNA, k , equals:

$$P = kM / (k_p + g)$$

First we assume that $k_p = 0$, the authors do not state this, I believe, but they do assume this. When M and k are growth rate independent then P should be inversely proportional to g : $P \propto 1/g$. In figure 2A, they show this dependence for GFP (they associate the concentration of GFP with GFP/OD, assuming that volume is proportional to OD) and P turns out to be proportional to g^{-3} instead of g^{-1} , to make this more clear I propose that the authors plot the GFP-growth rate loglinear and indicate a slope of -1 to emphasise the stronger dependency. So from this they conclude that k , M or both have g dependency: so,

$$P(g) = k(g)M(g)/g,$$

Since $M = w(g)/(k_m + g)$ with $w(g)$ as the transcription rate, which can be growth-rate dependent, and k_m as the mRNA degradation rate constant. The relation for M will generally simplify to $M = w(g)/k_m$, because generally $g \ll k_m$. Then the authors tested the dependency of M on growth rate, such a dependency was found to exist, in agreement with earlier findings by Gerosa et al in 2013. The dependence of M on g was however not sufficient to explain the dependence of P on g and, therefore, the translation efficiency, k , also depends on g . This k - g relation was deduced and plotted in figure 2C. The translation efficiency was concluded to decrease with growth rate. Next, the authors varied the translation-initiation-region (TIR) sequence of the GFP-transcript to address the growth rate dependence of GFP/OD varies with the TIR sequence and this indeed occurred. Different TIR sequences have a different growth rate dependency (Figure 3). Since the TIR sequence is involved in ribosome subunit binding, functional ribosome assembly, and initiation of translation elongation, the authors make a model of translation initiation, consider this model at quasi-steady state, and relate the above defined translation efficiency, k , to the kinetic and concentration parameters of this translation-initiation model. This leads to $k = K_1 R_f / (K_2 + R_f)$, so K_1 equals the maximal translation initiation rate in ribosomes/(min mRNA) = proteins/(min mRNA) and the K_2 as a Michaelis-Menten constant with as unit the free ribosome concentration. In this model it is assumed that the 30S-IFs-tRNA complex (X) is the limiting factor for translation initiation and not 50S (or other factors) or both. In the latter case, k would have equal to, in the simplest case, $K_1 [X] / (K_2 + [X]) [50S] / (K_3 + [50S])$. Also this assumption is not mentioned explicitly. With this model, the authors then illustrate that transcripts, with different TIRs, have different values for K_1 and K_2 such that the dependencies of their translation efficiencies on the free ribosome concentration are different (Figure 4). Next they use the GFP/OD data of the different TIR-constructs to fit for every TIR construct a K_1 and K_2 value and for all those data the same R_f /growth-rate dependence. Here we find a major weakness of the paper: since the limiting factor for translation initiation does not need to be the free ribosome, but can equally well be the 50S concentration, or the concentration of another translation-associated protein, the identity of the limiting factor, of which the concentration is plotted in Figure 5A, remains unclear. So they cannot argue that it must be the free ribosome concentration - although I agree that this is tempting, it is not a direct observation, only a deduction from a model, which has certain assumptions. Next they work no longer with synthetic constructs but with *B subtilis* mRNA and proteome data, use the fitted ribosome/growth-rate relation and fit the K_1 and K_2 parameters for 1002 transcripts of *B subtilis*. This leads to three classes of transcripts: transcripts that are R_f independent (R_f -saturated), nonlinearly dependent on R_f (R_f -unsaturated) and linearly dependent on R_f (R_f -undersaturated). This leads to several questions not addressed in this paper:

1. do the transcripts, that fall in those classes, have particular TIR similarities and differences, and are their translation-efficiency correlations only explained by TIR-sequence similarities or also by similarities in their protein-coding sequences?
2. in the figure 6 data, the transcript protein-coding sequences are no longer identical - in the previous experiment it was always the same GFP-transcript with a different TIR - such that translation-elongation regulation, via ribosome-pausing, ribosome-stalling/collisions, translation-termination regulation, etc, can also play a role, if those regulations are growth-rate dependent then it is not the free ribosome concentration that explains all translation efficiencies for those 1002 *B subtilis* transcript. Since this is the main point of the paper, I find that this claim is not sufficiently supported by the data. More efforts are required to substantiate the main claim given the data in Figure 6.

Next, the authors undertake a translation-inhibition experiment, with tetracycline, to show - I guess that this is what they hoped to get of this experiment, I do not understand their reasoning very well, so I would appreciate that this section is rewritten - that indeed the TIR sequence effect (or the free

ribosome concentration effect) is dominant over growth-rate-dependent elongation effects; they find that tetracycline effects are qualitatively growth-rate independent, suggesting that the TIR differences explain the growth rate dependency of translation efficiency in *B subtilis*. I do not find this an experiment that is well explained, so I may miss interpret its purpose. Finally, in Figure 8, the central mechanism, argued for in this work, is explained in a nutshell.

Minor comments:

1. the work abundance is often used, but it is unclear what units are meant, please address this, especially in the introduction and the discussion,
2. the growth rate (symbol μ) is not defined in the equation used in the introduction,
3. In line 6, page 4, in the sentence starting with "The comparison of these proteome", this statement needs a reference.
4. Page 5, line 16, the word "infirm" sounds a bit awkward to me, I would write validate/falsify instead of confirm/infirm.
5. Page 6, line 17, *dhaS* is mentioned but not shown in Figure 1F there *gcaD* is plotted.
6. page 8, line 16-18, please double check the TIR notations we have the impression that some errors occurred in the names, they do not all correspond to the names mentioned the main-text and supplemental figures.
7. Page 8, line 23, rewrite "combined experimental-computational dedicated approach"
8. Page 9, to me translational-efficiency trajectories sounds very awkward, I would swap trajectories for dependencies or relations.
9. Page 10, line 1, "for steady-state growth" appears out of place.
- 10 Page 12, line 2 and 3, this sentence does not run properly.
- 11 Page 14, line 10, "trigger a higher titration" sounds awkward please revise.
- 12 Page 15, line 2, "was not sufficient alone" rearrange
- 13 Page 15, line 17 and 18, "to add to the growth medium translation inhibitors" rearrange
- 14 Page 20, line 10, this sentence does not run properly.

1st Revision - authors' response

18 February 2016

Reviewer #1:

I believe that the findings that the authors made are of importance and this could at some point become a contribution for MSB. However, at its current state, the manuscript is not yet suitable for MSB. The text and the analyses are so convoluted that it is very difficult for a reader to understand and appreciate the work. Even a referee that really made an attempt and spent numerous hours on reviewing could not really appreciate (and evaluate!) the work.

- ⇒ We are very grateful to Reviewer #1 for its constructive comments and advices and to acknowledge that the findings are of importance and that this work could at some point become a contribution for MSB. In particular, we thank him/her for the time spent in reviewing the manuscript, which we concede in light of the comments was not rigorously written.

I strongly advise the authors to rewrite their manuscript, to adhere to the good practice of presenting ONE idea in ONE paragraph (not having paragraphs that go over more than one page) and to adopt a strict and logical structure.

- ⇒ We reorganized the structure of the paper to adopt a more logical structure and rewrote several parts following the recommendations of the three reviewers.

Below, please find a list with points that can serve as an indication for the things that prevented me from evaluating the work in depth. In my opinion the authors do not only need to address these points, but they need to completely rewrite the manuscript so that it can be evaluated.

- ⇒ We carefully tried to address each point in the new version. As you will see, thanks to the manuscript reorganization, some of your comments and critics have been implicitly addressed.

Next to this point, the authors also need to check one critical assumption that they made: p.9, supplementary text 2.1: The authors assume that dilution due to growth is much higher than protein degradation. While this might be correct for the high growth rates where this was probably tested, the question is whether this is also the case at lower growth rates. Given the importance of this assumption for the overall conclusion of this work, I feel that the authors need to experimentally test this assumption.

- ⇒ We fully agree that protein degradation is a critical issue to be considered. This is why among other reasons, we made use of the *gfpmut3* gene as a reporter gene. Indeed, we have previously shown that this GFP variant possesses a half-life over 20 hours in both poor and rich media (References from the main text: Botella et al. 2010; Buscher et al. 2012). In the former main text of the manuscript, this information was missing. In the new version, we therefore inserted a comment on page 6 lines 8-9.
- ⇒ When the model was applied to the whole proteome using proteomic datasets, we assumed that protein degradation was negligible since ribosomal and cytosolic proteins were previously found to be stable during exponential growth in Gram-negative and Gram-positive bacteria (Appendix §2.3; References from Appendix: Gur et al, 2011; Jayapal et al, 2010; Kock et al, 2004; Piir et al, 2011). Although protein degradation in exponentially growing *B. subtilis* cells is negligible according to the current knowledge, we cannot exclude (and we even expect) that a few proteins will specifically be targeted by growth-condition-specific post-transcriptional mechanisms and/or sent to the proteasome. In the new version and in line with this issue, we added a comment on page 10 lines 20-21.

First result paragraph; long, weird logic ... mix of method description, speculation, results, ... make a clear structure. It took me very long to get it. In combination with Units (relative, normalized), inconsistent use of terms make things very confusing; in Fig 1D: on the two y-axes two different units.

- ⇒ We agree with Reviewer #1. We honestly apologize to have left so many inconsistencies that led to a confusing picture of the results and we know as reviewers that it is might be very upsetting to read.
- ⇒ We simplified the first paragraph and removed speculations such as: '*Assuming that the proportion of B. subtilis tRNA abundance in total RNA is nearly constant in any growth conditions as it is for Escherichia coli (Bremer & Dennis, 2008; Kjeldgaard & Kurland, 1963; Maaløe & Kjeldgaard, 1966), the deduced total mRNA abundance in B. subtilis (TotmRNA=TotRNA-TottRNA) increases as fast as the growth rate does. If such is the case, the total mRNA abundance in B. subtilis increases twice faster than total mRNA abundance in E. coli with increasing growth rate (Bremer & Dennis, 2008; Maaløe & Kjeldgaard, 1966; Marr, 1991; Schaechter et al, 1958; Valgepea et al, 2013). To experimentally confirm or infirm this deduction, we quantified*'
- ⇒ Figure 1 was largely modified and simplified (and a some panels have been moved to Figure EV1).
- ⇒ We also removed the part corresponding to the methodology that was already explained in the Materials and Methods: '*Briefly, a defined amount of ten different in-vitro synthesized (spike-in) transcripts was added to an equal amount of each total RNA sample prior synthesis and hybridization of Cy3 fluorescently labeled cDNA to genome-wide microarrays (Figure 1B).*'
- ⇒ Overall we simplified the paragraph and homogenized the different units across all the figures of the manuscript.

From Fig S1A it seems that the authors have measured at different growth rates than the data shown in Fig 1A seems to indicate? Were these the same experiments? I would at least expect in Fig 1A to also see horizontal error bars.

- ⇒ This is our mistake. Reviewer #1 is right, we corrected the growth rates accordingly. Note that we also corrected this mistake in some other figures.
- ⇒ We included horizontal error bars. See new Figure EV1.

Fig S1B: The description on y-axis is confusing. If you say "rRNA distribution (%)" I expect values to sum up to 100%.

- ⇒ Figure S1 was completely redrew taking into account the different comments of each reviewer (see new Figure EV1C and D). The rRNA abundances were however kept as a percent of the total RNA abundance (for a sum of about 85%) to facilitate understanding of Figure EV1B.

p.5: The authors assume on page 5 around lines 9ff that the proportion of tRNA on total RNA is as constant over growth rates as it is E. coli. I find this a critical assumption, particularly because they show in Fig 1A that the total RNA abundance in B. subtilis behaves completely different across growth rates as in E. coli. I am particularly concerned that the authors than state in line 14/15 that the mRNA abundance will increase faster in Bsub than in Ecoli. I would simply remove this whole speculation (i.e. lines 9 to 16) and directly go and measure it.

- ⇒ We agree with Reviewer #1's comment. We removed the speculation (i.e. lines 9 to 16) since it is not related to the present work and does not affect the conclusions. In addition, we do not need in the present work to quantify total tRNA and total rRNA. We only need to know the proportion of total mRNA within total RNA. As mentioned earlier, we modified and simplified the first paragraph of the results. We also removed *E. coli* from Figure 1 since it is misleading and as highlighted by Reviewer #1 in a following critic, it is not rigorous to present total RNA by optical density units from different organisms.
- ⇒ All data were acquired relatively to the OD₆₀₀. Therefore, each of the estimation, calculation and computation performed are always ratios between quantities related to the same OD₆₀₀ values. For instance, the translation efficiency (λ_i) is proportional to the ratio between protein and mRNA abundances (implicitly each divided by OD₆₀₀), which implies that any coefficient related to the OD₆₀₀, the volume or the shape of the cells, *etc.* is included within the two measures and disappears after computation.

Description of the data shown in Fig 1F: The text is completely decoupled from what is shown in the figure; genes are mentioned (dhaS that are not shown in the figure; how can I see 2x more enrichment of fbaA, "at the boundary of the two classes"?)

- ⇒ We completely reorganized and simplified this paragraph. The two comments were then addressed. Please see the new version (p. 5).

Comments on Figure 1 (note similar things would apply to other figures as well): Some of the data is normalized to OD. And such normalized data is compared with respectively normalized data from E. coli. I find this problematic as ODs can be measured at different wavelength, even when measured at the same wavelength different spectrophotometers can give different readings, and finally growth-rate dependent cell morphology differences can also give different OD readings. Thus, in this quantitative work, I feel the authors should use more accurate normalizations.

- ⇒ As mentioned earlier, we have removed the part concerning *E. coli* since it does not bring any new idea and information and since as underlined by Reviewer #1, such a comparison is not rigorous. We next modified the figure and indicated that the y-axis was in normalized absorbance units for *B. subtilis* since we converted the data obtained in the group of Uwe Sauer (Dauner et al. 2001) in the same unit as we used for optical density. This was possible because our two labs already worked together using different media and normalized their results (Buescher et al., Science, 2012).

Fig. 1A: - add "total" to the y-axis

- ⇒ Checked and modified.

Fig. 1D: add "total" in front of "RNA" on the y-axis, what does "uOD" mean (i.e. the "u")? Why are there different units on the 2 y-axes. How can I then compare the different values?

Fig. 1C: Description of y-axis unclear

Fig. 1F: Not clear from the y-axis description what the data is normalized to?

Fig. 1E: mue2 is indicated to be 1.6, while in the text it is stated that the growth rates measured were 1.7. Furthermore, the shown equation is never mentioned and it is not clear to me what it is supposed to show me.

- ⇒ We agree, there were again inconsistencies between the unit and name of the y-axis.
- ⇒ The Figure 1 has been fully redrawn and the legend rewritten.
- ⇒ Units have been homogenized across figures.
- ⇒ We corrected axis descriptions.
- ⇒ We corrected the value 1.6 by 1.7h^{-1} .
- ⇒ The equation has been removed since we do not discuss it in the manuscript, it was only confusing.

Note that the figures in Fig 2 carry different units for the OD, indicating wavelength and U.OD instead of uOD.

- ⇒ Figure 2 was corrected accordingly to standardize the units notations across figures.

P.7, line 1: start with defining translation efficiency and motivate it

- ⇒ We now started the paragraph (new version page 6 line 4) with the following sentence: *'In this work, we aimed at experimentally determining the transcript-specific translation efficiency across growth conditions. The transcript-specific translation efficiency (λ_i in h^{-1}) is defined as the number of proteins produced per mRNA per hour, i.e. $\lambda_i = \frac{\mu P_i}{m_i}$ (Bremer & Dennis, 2008).'*

p.7. Line 10 "combining total mRNA quantification and qPCR-corrected transcriptome data I am lost. I am confused: Because in this sentence it seems that the determine gfpmut3 transcript abundance, but a few lines earlier they already mentioned that they did it and even showed a comparison in Fig 2B....

- ⇒ Reviewer #1 is right and we feel sorry again to have left such inconsistencies that are the results of several rearrangements of the manuscript. We removed the inconsistent part of the sentence: *'inferred the gfpmut3 mRNA abundance at various growth rates. We'*.
- ⇒ We rephrased and clarified the paragraph (page 6).

p.7 line 12: They then "deduced the translation efficiency". The authors have neither told the reader how they have defined translation efficiency nor did they tell the reader how they calculated. Also because there are no units on the y-axis of Fig 2C, I cannot deduce it myself. (It is only hidden in the caption of supplementary Figure 2).

- ⇒ We now defined translation efficiency at the beginning of the paragraph (p.5 lines 5-6). We also corrected Figure 2 accordingly.

p.7. lines 13-17, the mentioned data is not shown; further the authors don't provide any motivation for this experiment

- ⇒ We agree with Reviewer #1. We removed this paragraph and figure. The data are anyway later used when characterizing the library of TIR variants and provided in the Dataset EV files. The end of the paragraph was corrected accordingly.

Confusing: Is ribosomal binding sites the same as translational initiation regions?

- ⇒ RBS and TIR are not the same. RBS is a subpart of TIR. The text mentioned by Reviewer #1 is indeed incorrect. We replaced *'ribosomal binding sites'* by *'translational initiation regions'* in the paragraph. (New version page 6)

P8. Line 9: it sounds the authors have generated new strains, but in fact the same strains were mentioned a page earlier

- ⇒ Right, the paragraph was unclear. We entirely rephrased the paragraph (see page 7)

P8. Line 11: The way this sentences is phrased it sounds like the authors did something, but in fact the refer to a reference. This is incorrect use of a citation.

- ⇒ This was corrected when addressing the previous comment (see page 7).

P8. Line 18-20: Here, the author mention that they do a correction and then refer to the panel of supplementary Fig 3A-D. Here data is shown, but in no way explained how the correction was done. The reader is left with detective and inference work to find out what the authors meant and did.

Note, at this point of the manuscript I am already 2 hours busy in doing this detective work, but I still do not really understand it.

- ⇒ This was also rephrased when addressing the previous comment (see page 7). We really feel sorry for the hard time spent by Reviewer #1.

p.8 lines 21-25, p.9. lines 1-14: First a computational analysis is mention in the paragraph, then they jump to experimental data and draw some new conclusions from this data. Where is the logic? Why this jumping??? The computational analysis mention refers to obtaining statistical confidence, but I see no where any statistical significance measure - neither in the text nor in the figure Fif 3B.

- ⇒ Right, this was indeed weird. We reorganized and rewrote the paragraph.

p.9 line 19: The authors mention 3 steps and refer to Fig 4A, where 4 (!) steps are shown. Furthermore, the figure also shows the 50S unit, which is in turn not mentioned in the text. All of this is confusing.

- ⇒ The fourth step was removed from the figure. We also rephrased the corresponding text to clarify the model.
- ⇒ Note however that this part has been moved to the end of the manuscript since following the comments of Reviewer #3 we needed to reorganize the whole results part.

Caption of Fig 4A, the mention the rate of elongation (k_{pi}). First, k_{pi} is a parameter and not a rate. Second, what is the mechanistic basis of its growth rate dependence?

- ⇒ We agree. k_{pi} is equal to the inverse the population averaged time of protein elongation. We therefore corrected this in the new version (Text, Tables, Figures).
- ⇒ As mentioned in Appendix, it has been shown in bacteria that the rate of elongation is growth rate dependent (Bremer et Denis, 2008), and consequently k_{pi} too. However, there is no experimental indication about the mechanistic basis of its growth rate dependence, but it was suggested that the growth rate dependent abundances of each tRNA actually affects the rate of elongation (Marr et al, 1991).

p.11. Lines 3-4: They conclude that modeling translation as an elementary three step initiation process can reproduce the data. However, on the page earlier they made a statement that a reversibility is crucial to achieve this. What is now correct?

- ⇒ The paragraph has been clarified. Both reversible and irreversible reactions are essential to obtain a Michaelis-Menten-type rate law as in the 3 step translation initiation process.

p.11 Lines 7-8: Where can I see this? How can I understand this?

p.11, lines 10-11. Not clear how this was done. What was optimized?

- ⇒ We believe that the reorganization of the results part in the new version addressed these two issues.

p.11, lines 19: The authors talk about sensitivities and refer to Fig 5C but here I cannot see sensitivities. After some thinking, I think the authors mean that variations in parameter values can change the growth-rate dependencies of the translational efficiency.... But why the heck don't the authors simply say this and spare the reader some unnecessary thinking...?

- ⇒ Right! Thank you. Checked and corrected.

p.12, lines 13-14: It is completely unclear how I should derive the conclusion mentioned here from the Fig. 6A.

- ⇒ We followed the suggestion of Reviewer #3 to rephrase this paragraph. Please see the new version.

p.12, This paragraph is again very convoluted and the authors jump between different things: First, they mention that they wanted to estimate the parameters; then they make a statement on nonlinearity of certain transcripts' translation efficiency, and in the next sentence they make a statement that the two K parameters are correlated, and finally in the next sentence they make a yet unrelated statement on two specific transcripts. Either I don't get it at all, or this is very confusing.

- ⇒ We simplified and rewrote the paragraph (p. 9-10).

- ⇒ We also removed the sentences concerning the two transcripts, since this was, we concede, very confusing in the paragraph.

p.12/13. After the authors have mentioned Fig. 6A in their text, they jump to Fig.8 leaving out the rest of the figures in Fig 6 and the Fig 7. This is very unprofessional and very confusing.

- ⇒ We fully agree and apologize. This was corrected in the new version.

p.13. lines 10-17: I am completely lost. Are the authors again jumping?

- ⇒ We split the paragraph and simplified the text. Please see the new version of the results part.

p.13, line 24: What is a graphical analysis? What was done here? What do they mean by an "event"?

- ⇒ Graphical analysis has been removed and rephrased.
 ⇒ By event, we meant a possible scenario that allowed exploring a model and the consequences of the variation of one parameter. We replaced 'event' by 'solution' and rephrased the paragraph (page 13).

p.14, line 7: what is a global reorganization of mRNA synthesis?

- ⇒ This part was modified following the denomination proposed by Reviewer #3 (R_{free} saturated, etc.), which we believe clarify the paragraph (from page 9 line20 to page 10 line 5)

p.14: What is described in this section would rather belong to the discussion.

- ⇒ Indeed, we simplified the paragraph. However, we consider that the analysis of the mathematical model is a result although it is not in the sense of experimental biology. In addition, the conclusion of the paragraph and of the results part is in our view of particular interest (page 14 lines 8-13).

p. 14, lines 16-17. Where can this be seen? What is "significant"?

- ⇒ Right. 'Significant' was replaced by 'important'

p.14, lines 17-20. Where can this be seen?

- ⇒ We now referred to the Dataset EV3 in which we highlighted the 32 ribosomal proteins (p. 13 line 25).

⇒

Reviewer #2:

*This manuscript addresses growth rate dependencies of protein and mRNA concentrations as well as their ratio, the translation efficiencies in *B. subtilis*. It reports the unexpected result that translation of individual mRNAs changes differentially as a function of growth rate, which is interpreted as reflecting a changing concentration of free ribosomes and mRNAs with ribosome binding sites that have different parameters in a Michaelis-Menten type model. These results are obtained by a nice combination of transcriptome and proteome wide data, analysis of gfp constructs with mutated ribosome binding sites and mathematical modeling. In my opinion, the manuscript is very well suited for publication in Mol Sys Biol.*

I have reviewed a previous version of this manuscript for another journal. Since then the manuscript has been substantially modified and extended. In particular, the concerns I had with the introduction and the discussion of the previous version have been resolved by the revision.

- ⇒ We are grateful to Reviewer #2 for his nice and constructive comments.

However, I still have two issues with this manuscript.

It is not clear to me how the elongation of the peptide chain enters the analysis of the proteome data. In the model, elongation is included in the rate $k_{p,i}$, which should depend both on the speed of elongation (and thus on codon usage, tRNA concentrations etc) and on the length of the transcript. The authors state in the supporting information that this parameter does not affect the Michaelis-Menten behavior of the translation rate and I agree with that statement as long as R_{free} is considered as known. However, for the global model, where R_{free} is also unknown, the values of

the $k_{p,i}$ will affect R_{free} . Thus, I think $k_{p,i}$ values should somewhere be obtained from the proteome analysis and I do not see where. In my opinion, this points needs an explanation. As a consequence, some properties attributed to ribosome binding sites could reflect sequences properties further downstream.

- ⇒ Overall, Reviewer #2 is right when saying that $k_{p,i}$ should depend both on the speed of elongation (and thus on codon usage, tRNA concentrations etc) and on the length of the transcript. However, we believe that the previous version of the manuscript led to some confusing interpretation. Please read the new version of the manuscript (the reorganization followed several comments formulated by Reviewer #3 in line with this comment).
- ⇒ We reorganized the results part and we believe that it addressed this comment. Indeed k_{pi} was not used to analyze the proteome dataset that is why we directly inferred R_{free} as a global parameter.
- ⇒ In the last paragraph of the manuscript we theoretically extended our model to the entire proteome but we did not make use of the proteome dataset. This paragraph is only a mathematical analysis of the different solutions that can explain the drop in *free* ribosome abundance. Hence, we drew general conclusion without accessing the values of k_{pi} .

*A minor issue is whether the increase of ribosome concentration upon growth in tetracycline has been tested. I think this is a necessary control, because the paper of Scott et al. reported that effect in *E. coli* and the control of ribosome synthesis is (partly) different in *B. subtilis* than in *E. coli*.*

- ⇒ This comment has been addressed with the reorganization of the results part. We also clarified this part.
- ⇒ We actually did not test the increase of ribosome abundance upon addition of tetracycline since we only needed to disturb the translation process (whatever the underlying mechanism) and check whether the translation efficiency has been affected with respect to the rate of growth. Hence, we concluded that there is no strict control mechanism that can rectify protein production with respect to the rate of growth (page 11 lines 20-23).

Reviewer #3:

*The authors report an interesting study on the dependency of translation (initiation) efficiency/rate - the translation rate per mRNA - on growth rate in *Bacillus subtilis*. They argue that:*

- 1. the translation-efficiency/growth-rate dependency in *B subtilis* is transcript specific and arises via a non-regulated, passive phenomenon,*
- 2. the mechanistic origin of this dependency derives from the reduction of the free ribosome concentration with growth rate, which affects translation initiation rate,*
- 3. the growth-rate dependence of the translation efficiency of transcripts varies between transcripts because their translation-efficiencies have a different dependence on the free ribosome concentration,*
- 4. The free-ribosome-concentration dependence of the translation efficiency of different transcripts varies when their translation-initiation-region sequences vary, giving rise to different translation-initiation kinetics and a different dependency of the initiation rate with the free ribosome concentration.*

- ⇒ We are very grateful to Reviewer #3 for his constructive advices that led us to fully reorganize the results part and we believe strongly improved the overall manuscript.
- ⇒ After a detailed reading of the comments and critics of Reviewer #3, we understand that a part of his comments are probably due to an inappropriate organization of the results part. Indeed, the introduction of the Michaelis-Menten-type rate law as a consequence of the translation initiation process at the molecular level and the use of a (large) set of assumptions led to introduce important confusions on the paper contributions. The way used to introduce the model clearly led Reviewer #3, to make several comments where he pointed out the weakness of our results in view of the (large) set of assumptions which could not be (experimentally) validated.
- ⇒ Actually, the main contribution of the paper is to experimentally show that the translation efficiency of messengers can be accurately estimated with respect to the growth rate on the basis of the use of a Michaelis-Menten type rate law (with two parameters and a global,

common and growth rate dependent entity, the so-called *free* ribosome). Our experimental efforts were mainly to prove that this empirical model allows to accurately fit a library of well-defined GFP reporter strains.

- ⇒ The other contributions of this work were dedicated to analyze/explore the consequences of the model at the genome scale and to get insights on the possible links between the proposed empirical model and the models which can be built and simplified on the basis of the current available knowledge about translation.
- ⇒ Consequently, since the organization of the results part was clearly inappropriate and introduced several confusions, we have drastically reorganized the manuscript.

The authors use experiments and modelling to illustrate and substantiate those claims.

I outline my general understanding of the theoretical concepts and experiments underlying this work one by one, I do this because the authors are sometimes a bit sloppy with their use of quantitative concepts, what their units are, and which modelling assumptions are made. For instance, the key concept of this work is the "translation efficiency", it is however used in two consecutive sentences in different ways: on page 3, line 23 and 24 the translation efficiency is defined as the number of proteins produced per mRNA, whereas in the next sentence, on page 4, it is defined as a rate, namely as the number of proteins produced per mRNA per unit time. I take the latter to be the correct definition. I propose that the authors define those concepts unambiguously with units and symbols, perhaps in a table or box.

- ⇒ Right! We corrected the definition as you suggested. We also included these concepts in the main text and the figure legends in order to avoid any ambiguity.

When I do this my understanding of the paper is as follows. At balanced growth, the concentration of proteins, P , is fixed such that its relation with its cognate mRNA concentration, M , the growth rate, g , the protein degradation rate, kp , and the translation rate per mRNA, k , equals:

$$P = kM / (kp + g)$$

- ⇒ Right!

First we assume that $kp=0$, the authors do not state this, I believe, but they do assume this.

- ⇒ Totally right, we do not explicitly mention the assumption but it is for sure a critical one. Please read below the answer given to Reviewer #1 which we pasted below:

- *We fully agree that protein degradation is a critical issue to be considered. This is why among other reasons, we made use of the *gfpmut3* gene as a reporter gene. Indeed, we have previously shown that this GFP variant possesses a half-life over 20 hours in both poor and rich media (References from the main text: Botella et al. 2010; Buscher et al. 2012). In the former main text of the manuscript, this information was missing. In the new version, we therefore inserted a comment on page 6 lines 8-9.*
- *When the model was applied to the whole proteome using proteomic datasets, we assumed that protein degradation was negligible since ribosomal and cytosolic proteins were previously found to be stable during exponential growth in Gram-negative and Gram-positive bacteria (Appendix §2.3; References from Appendix: Gur et al, 2011; Jayapal et al, 2010; Kock et al, 2004; Piir et al, 2011). Although protein degradation in exponentially growing *B. subtilis* cells is negligible according to the current knowledge, we cannot exclude (and we even expect) that a few proteins will specifically be targeted by growth-condition-specific post-transcriptional mechanisms and/or sent to the proteasome. In the new version, we therefore inserted a sentence on page 10 lines 20-21.*

When M and k are growth rate independent then P should be inversely proportional to g : $P \propto 1/g$.

- ⇒ Right!

In figure 2A, they show this dependence for GFP (they associate the concentration of GFP with GFP/OD, assuming that volume is proportional to OD) and P turns out to be proportional to g^{-3}

instead of g^{-1} , to make this more clear I propose that the authors plot the GFP-growth rate loglinear and indicate a slope of -1 to emphasise the stronger dependency.

- ⇒ We agree with the Reviewer #3 suggestion. We included two very nice figures as proposed (Figures EV2E-F, see below) using the datasets from the Figure 2 and Figure 3. Figure EV2E and F were cited when required to emphasize the stronger dependency (page 7, line 18). Please see the figure legends of Figure EV2 panels E and F.
- ⇒ We did not wish to modify the figures from the main text since we believe it would be

harder for the readers to compare the different figure panels as for instance of Figure 2.

So from this they conclude that k , M or both have g dependency: so,

$$P(g) = k(g)M(g)/g,$$

- ⇒ Right!

Since $M = w(g)/(km + g)$ with $w(g)$ as the transcription rate, which can be growth-rate dependent, and km as the mRNA degradation rate constant. The relation for M will generally simplify to $M = w(g)/km$, because generally $g \ll km$.

- ⇒ Note that the degradation of the messengers also depends on the growth rate variations (see Figure EV2).

Then the authors tested the dependency of M on growth rate, such a dependency was found to exist, in agreement with earlier findings by Gerosa et al in 2013. The dependence of M on g was however not sufficient to explain the dependence of P on g and, therefore, the translation efficiency, k , also depends on g .

- ⇒ Right

This k - g relation was deduced and plotted in figure 2C. The translation efficiency was concluded to decrease with growth rate. Next, the authors varied the translation-initiation-region (TIR) sequence of the GFP-transcript to address the growth rate dependence of GFP/OD varies with the TIR sequence and this indeed occurred. Different TIR sequences have a different growth rate dependency (Figure 3). Since the TIR sequence is involved in ribosome subunit binding, functional ribosome assembly, and initiation of translation elongation, the authors make a model of translation initiation, consider this model at quasi-steady state, and relate the above defined translation efficiency, k , to the kinetic and concentration parameters of this translation-initiation model.

- ⇒ Right

This leads to $k = K1 Rf/(K2 + Rf)$, so $K1$ equals the maximal translation initiation rate in ribosomes/(min mRNA) = proteins/(min mRNA) and the $K2$ as a Michaelis-Menten constant with as unit the free ribosome concentration. In this model it is assumed that the 30S-IFs-tRNA complex (X) is the limiting factor for translation initiation and not 50S (or other factors) or both. In the latter case, k would have equal to, in the simplest case, $K1 [X]/(K2 + [X]) [50S]/(K3 + [50S])$.

- ⇒ We agree with this remark and that explains why we have explored the impact of 50S in section 2.4.1 of the Appendix (even if we do not discuss in this analysis the fact that the '50S' free ribosome can be the only 'key' limiting step). From a mathematical viewpoint, any components involved in the translation initiation process can be “the key player” and generally lead to a Michaelis-Menten type rate law.

Also this assumption is not mentioned explicitly. With this model, the authors then illustrate that transcripts, with different TIRs, have different values for K1 and K2 such that the dependencies of their translation efficiencies on the free ribosome concentration are different (Figure 4). Next they use the GFP/OD data of the different TIR-constructs to fit for every TIR construct a K1 and K2 value and for all those data the same Rf/growth-rate dependence. Here we find a major weakness of the paper: since the limiting factor for translation initiation does not need to be the free ribosome, but can equally well be the 50S concentration, or the concentration of another translation-associated protein, the identity of the limiting factor, of which the concentration is plotted in Figure 5A, remains unclear.

- ⇒ See the comments related to reorganization of the results part.

So they cannot argue that it must be the free ribosome concentration - although I agree that this is tempting, it is not a direct observation, only a deduction from a model, which has certain assumptions. Next they work no longer with synthetic constructs but with B subtilis mRNA and proteome data, use the fitted ribosome/growth-rate relation and fit the K1 and K2 parameters for 1002 transcripts of B subtilis. This leads to three classes of transcripts: transcripts that are Rf independent (Rf-saturated), nonlinearly dependent on Rf (Rf-unsaturated) and linearly dependent on Rf (Rf-undersaturated).

- ⇒ We are grateful to Reviewer #3 for this particular comment. Indeed, we decided to use the denomination 'Rf-saturated, Rf-unsaturated and linearly Rf-undersaturated transcripts' in the main text and the figures since we believe it will highly ease the reading of the corresponding paragraphs (from page 9 line 20 to page 10 line 5).

This leads to several questions not addressed in this paper:

1. *do the transcripts, that fall in those classes, have particular TIR similarities and differences, and are their translation-efficiency correlations only explained by TIR-sequence similarities or also by similarities in their protein-coding sequences?*

⇒ We tried to explore this point but a key difficulty in this context is mainly related to the fact that the links between the sequences of the TIRs and “their properties” remain very elusive. The properties of few TIR have been studied in detail. For example, we tried to confirm the fact that the length of the 5'UTR and the parameter K_{2i} are related, however, we did not find any clear trend.

⇒ Obviously investigating the TIR properties genome-scale remains a challenging issue that should be addressed in future works.
2. *in the figure 6 data, the transcript protein-coding sequences are no longer identical - in the previous experiment it was always the same GFP-transcript with a different TIR - such that translation-elongation regulation, via ribosome-pausing, ribosome-stalling/collisions, translation-termination regulation, etc, can also play a role, if those regulations are growth-rate dependent then it is not the free ribosome concentration that explains all translation efficiencies for those 1002 B subtilis transcript. Since this is the main point of the paper, I find that this claim is not sufficiently supported by the data. More efforts are required to substantiate the main claim given the data in Figure 6.*

⇒ As explained in a previous comment, the main contribution of the paper is to show that a Michaelis-Menten type rate law accurately fit translation efficiencies with respect to the growth rate. Even if we made a quite theoretical study to get insights on the molecular composition of Rfree and on its growth-rate dependent variation, we do not claim that Rfree can be predicted but only estimated from genome-wide datasets.

- ⇒ Actually, we believe that this comment is the consequence of the inappropriate organization of the results part.

*Next, the authors undertake a translation-inhibition experiment, with tetracycline, to show - I guess that this is what they hoped to get of this experiment, I do not understand their reasoning very well, so I would appreciate that this section is rewritten - that indeed the TIR sequence effect (or the free ribosome concentration effect) is dominant over growth-rate-dependent elongation effects; they find that tetracycline effects are qualitatively growth-rate independent, suggesting that the TIR differences explain the growth rate dependency of translation efficiency in *B subtilis*. I do not find this an experiment that is well explained, so I may miss interpret its purpose.*

- ⇒ We clarified and simplified this paragraph (see page 11)

Finally, in Figure 8, the central mechanism, argued for in this work, is explained in a nutshell.

- ⇒ We are not sure to understand this comment. Perhaps, this comment again is the consequence of our confusing presentation of the study contribution. We anyway think that this comment has been addressed in the new version.

Minor comments:

1. the work abundance is often used, but it is unclear what units are meant, please address this, especially in the introduction and the discussion,

- ⇒ we addressed this issue by modifying figure legends and some part of the main text. We hope this not an issue anymore in the new version

2. the growth rate (symbol μ) is not defined in the equation used in the introduction,

- ⇒ Corrected (page 4, line 1)

3. In line 6, page 4, in the sentence starting with "The comparison of these proteome", this statement needs a reference.

- ⇒ We do not know references stating this across growth conditions. This statement comes from personal communications and analysis following Muntel et al. 2014 (referred in the manuscript) between V. Fromion, M. Jules, U. Mäder and D. Becher.
⇒ If allowed by the editorial board, we may cite it this way in the introduction part.

4. Page 5, line 16, the word "infirm" sounds a bit awkward to me, I would write validate/falsify instead of confirm/infirm.

- ⇒ The paragraph has been removed following advices from Reviewer #1.

5. Page 6, line 17, dhaS is mentioned but not shown in Figure 1F there gcaD is plotted.

- ⇒ The sentence was removed following advices from Reviewer #1.

6. page 8, line 16-18, please double check the TIR notations we have the impression that some errors occurred in the names, they do not all correspond to the names mentioned the main-text and supplemental figures.

- ⇒ We carefully checked this point but did not find the errors. Could you precise which names were perhaps misspelled?

7. Page 8, line 23, rewrite "combined experimental-computational dedicated approach"

- ⇒ The paragraph was deeply modified (page 7 lines 15-22).

8. Page 9, to me translational-efficiency trajectories sounds very awkward, I would swap trajectories for dependencies or relations.

- ⇒ We are not very happy with 'trajectories' but we were not convinced by any other words. Actually we would like to keep the concept of 'growth-rate-dependency of the translation efficiencies' and therefore the word 'dependencies' did not sound appropriate for us.
⇒ If required, we will rephrase it.

9. Page 10, line 1, "for steady-state growth" appears out of place.

- ⇒ Right! This was removed.

10 Page 12, line 2 and 3, this sentence does not run properly.

⇒ We removed this sentence and modified the introduction of the following paragraph (page 9 lines 9-14).

11 Page 14, line 10, "trigger a higher titration" sounds awkward please revise.

⇒ The sentence was rephrased (page 13 line 20).

12 Page 15, line 2, "was not sufficient alone" rearrange

⇒ The sentence was rephrased (page 14 line 7).

13 Page 15, line 17 and 18, "to add to the growth medium translation inhibitors" rearrange

⇒ The sentence was rephrased (page 11 line 7).

14 Page 20, line 10, this sentence does not run properly.

⇒ The paragraph was deeply modified (page 18 line 10).

2nd Editorial Decision

15 April 2016

Thank you again for submitting your work to Molecular Systems Biology. First of all, I would like to apologize for the exceptional delay in getting back to you. Unfortunately after a series of reminders, we have not managed to obtain a report from Reviewer #3. As such, and in order to not delay the process any further, we have decided to make a decision based on the comments of Reviewer #2. As you will see below, this referee is overall satisfied with the revised study and only raises a couple of minor issues, which we would ask you to address in a revision.

Reviewer #2:

In my opinion, the paper has been further improved and the responses to the reviewers are clear and satisfactory. I recommend to accept the paper for publication, but have two remaining minor comments:

- 1) I agree with reviewer 3 in his/her previous report that the term "translation efficiency trajectories" is awkward. How about just calling this "growth-rate dependencies" or different "functional forms of the translation efficiency's growth rate dependence"?
- 2) I would recommend that the statements about evolutionary optimization in the last section of the discussion are formulated more carefully. In my opinion, the different growth rate dependencies provide opportunities for optimization, but based on present evidence no strong claims can be made.

2nd Revision - authors' response

18 April 2016

Reviewer #2:

In my opinion, the paper has been further improved and the responses to the reviewers are clear and satisfactory. I recommend to accept the paper for publication, but have two remaining minor comments:

⇒ We are very grateful to Reviewer #2 for his recommendation.

1) I agree with reviewer 3 in his/her previous report that the term "translation efficiency trajectories" is awkward. How about just calling this "growth-rate dependencies" or different "functional forms of the translation efficiency's growth rate dependence"?

As proposed by Reviewer #2, we simplified the manuscript and replaced the terms:

- ⇒ "translation efficiency trajectories" (p.6 l.23, p.9 l.10, p.9 l.13/14, p.15 l.23/24, p.17 l.2, p.17 l.5/6, p.17 l.9) and " μ -dependent translation efficiency trajectories" (p.11 l.22/23, p.17 l.22) by "translation efficiency's growth rate dependencies"
- ⇒ "translation efficiency trajectories across growth conditions" by "growth-rate dependencies" (p.7 l.4, p.18 l.13)
- ⇒ "followed various growth-rate-dependent trajectories" by "exhibited various growth-rate dependencies" (p.10 l.19)
- ⇒ "follow different growth-rate dependent trajectories" by "exhibit different growth-rate dependencies" (p.15 l.22/23, p.16 l.16)

2) I would recommend that the statements about evolutionary optimization in the last section of the discussion are formulated more carefully. In my opinion, the different growth rate dependencies provide opportunities for optimization, but based on present evidence no strong claims can be made. We agree with Reviewer #2 that the statements about evolutionary optimization in the last section of the discussion were too strong. We therefore replaced the terms "provides" (p.18 l.25) by "may provide", "allow" (p.19 l.1) by "may allow" as well as "contribute" (p.19 l.8) by "may contribute", and the word "contributes" (p.19 l.11) was removed.

Corresponding Author Name: Dr. Jules Matthieu & Dr. Vincent Fromion

Manuscript Number: MSB-15-6608